RESEARCH COMMUNICATION

# Reduced signal for polygenic adaptation of height in UK Biobank

**Jeremy J Berg[1†]\*, Arbel Harpak[1,2†], Nasa Sinnott-Armstrong[3†], Anja Moltke Joergensen[4], Hakhamanesh Mostafavi[1], Yair Field[3], Evan August Boyle[3], Xinjun Zhang[5], Fernando Racimo[4], Jonathan K Pritchard[2,6]\*, Graham Coop[7,8]\***

[1]Department of Biological Sciences, Columbia University, New York, United States; [2]Department of Biology, Stanford University, Stanford, United States; [3]Department of Genetics, Stanford University, Stanford, United States; [4]Lundbeck GeoGenetics Centre, Department of Biology, University of Copenhagen, Copenhagen, Denmark; [5]Department of Anthropology, University of California, Davis, Davis, United States; [6]Howard Hughes Medical Institute, Stanford University, Stanford, United States; [7]Center for Population Biology, University of California, Davis, Davis, United States; [8]Department of Evolution and Ecology, University of California, Davis, Davis, United States

**\*For correspondence:**
jeremy.jackson.berg@gmail.com (JJB);
pritch@stanford.edu (JKP);
gmcoop@ucdavis.edu (GC)

[†]These authors contributed equally to this work

**Competing interests:** The authors declare that no competing interests exist.

**Abstract** Several recent papers have reported strong signals of selection on European polygenic height scores. These analyses used height effect estimates from the GIANT consortium and replication studies. Here, we describe a new analysis based on the the UK Biobank (UKB), a large, independent dataset. We find that the signals of selection using UKB effect estimates are strongly attenuated or absent. We also provide evidence that previous analyses were confounded by population stratification. Therefore, the conclusion of strong polygenic adaptation now lacks support. Moreover, these discrepancies highlight (1) that methods for correcting for population stratification in GWAS may not always be sufficient for polygenic trait analyses, and (2) that claims of differences in polygenic scores between populations should be treated with caution until these issues are better understood.
**Editorial note:** This article has been through an editorial process in which the authors decide how to respond to the issues raised during peer review. The Reviewing Editor's assessment is that all the issues have been addressed (see decision letter).
DOI: https://doi.org/10.7554/eLife.39725.001

## Introduction

In recent years, there has been great progress in understanding the polygenic basis of a wide variety of complex traits. One significant development has been the advent of 'polygenic scores', which aim to predict the additive genetic component of individual phenotypes using a linear combination of allelic contributions to a given trait across many sites. An important application of polygenic scores has been the study of polygenic adaptation—the adaptive change of a phenotype through small allele frequency shifts at many sites that affect the phenotype.

Thus far, the clearest example of polygenic adaptation in humans has come from analyses of polygenic scores for height in Europe. However, as we will show here, this signal is strongly attenuated or absent using new data from the UK Biobank (*Bycroft et al., 2018*), calling this example into question.

Starting in 2012, a series of papers identified multiple lines of evidence suggesting that average polygenic scores for height increase from south-to-north across Europe (*Table 1*). Analyses from multiple groups have concluded that the steepness of this cline is inconsistent with a neutral model of evolution, suggesting that natural selection drove these differences in allele frequencies and polygenic scores (*Turchin et al., 2012*; *Berg and Coop, 2014*; *Robinson et al., 2015*; *Zoledziewska et al., 2015*; *Berg et al., 2017*; *Racimo et al., 2018*; *Guo et al., 2018*). Significant differences in polygenic scores for height have also been reported among ancient populations, and these are also argued to have been driven by selection (*Mathieson et al., 2015*; *Martiniano et al., 2017*; *Berg et al., 2017*). In parallel, (*Field et al., 2016*) developed the Singleton Density Score (SDS)—which compares the distance to the nearest singleton on two alternative allelic backgrounds—to infer recent changes in allele frequencies, and used it to analyze a large sample of British individuals (the UK10K; *UK10K Consortium, 2015*). They found a significant covariance of SDS and effect on height, suggesting that natural selection drove a concerted rise in the frequency of height-increasing alleles in the ancestors of modern British individuals during the last 2,000 years (*Field et al., 2016*).

All such studies rely on estimates of individual allelic effects on height, as calculated from genome-wide association studies (GWAS). These GWAS estimates are then combined with population-genetic analysis to test for selection. Under a null model of no directional change, we would not expect 'tall' alleles to increase (or decrease) in frequency in concert; thus, loosely speaking, a systematic shift in frequency of 'tall' alleles in the same direction has been interpreted as evidence for selection.

While our focus here is on the the distribution of height polygenic scores in Europe, we see this as a case study for understanding the challenges in comparing polygenic scores across populations in general. Compared to other complex traits, height is particularly well-characterized, and the

**Table 1.** Studies reporting signals of height adaptation in Europeans.

Prior to the UK Biobank dataset, studies consistently found evidence for polygenic adaptation of height. Notes: Most of the papers marked as having 'strong' signals report p-values $<10^{-5}$, and sometimes $<<10^{-5}$. In the present paper, the UK Biobank analyses generally yield p-values $>10^{-3}$.

| GWAS | Approach | Signal | Reference |
|---|---|---|---|
| GIANT 2010 | European frequency cline of top SNPs | strong | *Turchin et al., 2012* |
| validation: Framingham sibs | | | |
| GIANT 2010 | Polygenic measures of pop. frequency differences | strong | *Berg and Coop, 2014* |
| GIANT | Polygenic measures of pop. frequency differences | strong | *Berg et al., 2017* |
| | | strong | *Racimo et al., 2018* |
| | | strong | *Guo et al., 2018* |
| | Polygenic diffs between ancient and modern populations | strong | *Mathieson et al., 2015* |
| GIANT | Heterogeneity of polygenic scores among populations | strong | *Robinson et al., 2015* |
| validation: R15-sibs | | | |
| Sardinia cohort | Low polygenic height scores in Sardinians. Effect estimates from Sardinian cohort at GIANT hit SNPs | strong | *Zoledziewska et al., 2015* |
| GIANT and R15-sibs | Singleton density (SDS) in UK sample vs GWAS | strong | *Field et al., 2016* |
| | Also: LD Score regression (SDS vs GWAS) | strong | |
| UK Biobank | Population frequency differences | weak or absent | This paper* |
| | Singleton density (SDS) in UK sample | weak or absent | This paper* |
| | LD Score regression (SDS vs GWAS) | weak | This paper* |

*See also results from *Sohail et al., 2019*.

DOI: https://doi.org/10.7554/eLife.39725.002

evidence for adaptation of height in Europeans seemed clear. Thus our work highlights a need for caution in this area until these issues are more fully understood (*Novembre and Barton, 2018*).

## GWAS data used to study adaptation of height

Until recently, the largest height GWAS dataset came from the GIANT consortium (253,288 individuals as of 2014; *Wood et al., 2014*). This is the primary GWAS underlying most studies of adaptation of height. Additionally, several groups have used other, smaller, datasets to replicate signals found using GIANT (*Turchin et al., 2012*; *Robinson et al., 2015*; *Zoledziewska et al., 2015*; *Field et al., 2016*; *Berg et al., 2017*). In particular, because it is known that population structure may be a confounder in GWAS, leading to false positive inferences of polygenic adaptation, several groups sought to replicate signals using family-based analyses, which protect against confounding due to stratification (*Allison et al., 1999*; *Spielman and Ewens, 1998*; *Abecasis et al., 2000*).

The first replication, by *Turchin et al. (2012)*, showed that the effect sizes of the top 1,400 GIANT associations (based on an earlier version of GIANT, published by (*Lango Allen et al., 2010*) were statistically consistent with effect sizes re-estimated in a smaller sibling-based regression approach using data from the Framingham Heart Study (4819 individuals across 1761 nuclear sibships from *Splansky et al., 2007*). Sibling-based regression is considered to be immune to confounding by population structure, and so the agreement of effect sizes between studies was taken as validation of the north-south gradient observed when using the GIANT effect sizes.

The second, partially independent, replication came from *Zoledziewska et al. (2015)*, who selected 691 height-associated SNPs on the basis of the GIANT association study, and then computed polygenic scores using effect sizes re-estimated in a cohort of 6307 individuals of Sardinian ancestry. They determined that the average polygenic score of Sardinian individuals was significantly lower than observed for other European populations, consistent with the previously reported north-south gradient of polygenic scores.

A third replication was performed by *Robinson et al. (2015)*, who used a different, larger sibling-based GWAS to identify associations ($\sim 17,500$ sibling pairs from *Hemani et al., 2013*). We refer to this sibling-based dataset as 'R15-sibs'. These authors showed that the north-south frequency gradient replicates using SNPs ascertained from the sibling-based GWAS. This replication is stronger than that performed by either *Turchin et al. (2012)*, or that by *Zoledziewska et al. (2015)*, as the cohort is larger and the SNP ascertainment did not rely on GIANT. As pointed out in the supplementary note of *Robinson et al. (2015)*, this two-step procedure—ascertaining with a large but potentially biased GWAS like GIANT, before switching to a less powerful but hopefully unbiased replication GWAS—has the potential to introduce an ascertainment bias, even if the effects are correctly estimated in the replication study (we note that a small fraction of the GIANT samples are contained within the R15-sibs analysis, so the effect sizes are not strictly independent; however, because of the sibling design, any bias due to stratification in GIANT should be absent in R15-sibs). The R15-sibs study was also used by *Field et al. (2016)* to verify a signal of recent selection in ancestors of the present day British population. *Field et al. (2016)* found that the signal of selection was fully replicated when using R15-sibs data.

Lastly, *Field et al. (2016)* also used LD Score regression to test for height adaptation in the British while controlling for population structure (*Bulik-Sullivan et al., 2015a*; *Bulik-Sullivan et al., 2015b*). While LD Score regression is typically used to estimate genetic covariance between two phenotypes, *Field et al. (2016)* used it to test for a relationship between height effects and a recent increase in frequency (measured by SDS)—and found a strong covariance of the two consistent with selection driving allele frequency change at height loci.

Here, we reassess these previously reported signals using data from the UK Biobank (UKB) with genotype and phenotype data for nearly 500,000 residents of the United Kingdom (*Bycroft et al., 2018*). UKB has recently become a key resource for GWAS, thanks to its large sample, the relatively unstructured population (compared to international studies such as GIANT), and the opportunity for researchers to work directly with the genotype data rather than with summary statistics.

This paper has two aims. First, we will show that previously reported signals of directional selection on height in European populations generally do not replicate when using GWAS effect estimates from the UK Biobank. Similar findings have been obtained independently by other groups working in parallel (*Sohail et al., 2019*; *Uricchio et al., 2019*). Second, we will show that both the GIANT and R15-sibs GWAS are confounded due to stratification along the North-South gradient

where signals of selection were previously reported. Signals detected using R15-sibs effect estimates were previously used as a significant source of evidence in favor of adaptation by *Field et al. (2016)*, as well as in (*Berg et al., 2017*). However, the investigators leading the (*Robinson et al., 2016*) GWAS have now confirmed that the effect size estimates released from their 2015 study were strongly affected by population structure due to a computational bug (*Robinson and Visscher, 2018*). We include an analysis of the R15-sibs GWAS here to document how these spurious signals affected previous inferences, as well as the evidence that indicated the presence of confounding in the data prior to detection of the bug.

The conclusion that adaptation signals in GIANT were spurious has broader implications for GWAS analysis, as it indicates that standard approaches for population structure correction may not always be sufficient, and that further study is needed to understand their limitations. While we anticipate that current methods are likely adequate for many applications, in particular for identification and broad-scale localization of strong genotype-phenotype associations—they may be insufficient for applications such as phenotypic prediction and the detection of polygenic adaptation as these can be highly prone to the cumulative bias through uncorrected structure. Such analysis should be undertaken with great care.

## Results

### GWAS datasets

We downloaded or generated seven different height GWAS datasets, each relying on different subsets of individuals or using different analysis methods. The bold-faced text give the identifiers by which we will refer to each dataset throughout this paper. These include two previous datasets that show strong evidence for polygenic adaptation, as well as an updated version of the R15-sibs dataset released in response to results in the initial preprint version of this manuscript:

**GIANT**: ($n$ = 253 k) 2014 GIANT consortium meta-analysis of 79 separate GWAS for height in individuals of European ancestry, with each study independently controlling for population structure via the inclusion of principal components as covariates (*Wood et al., 2014*).

**R15-sibs**: ($n$ = 35 k) Family-based sib-pair analysis of data from European cohorts (*Hemani et al., 2013*; *Robinson et al., 2015*). The effect sizes associated with this paper were publicly released in 2016 (*Robinson et al., 2016*).

**R15-sibs-updated**: ($n$ = 35 k) In November 2018, while this paper was in the final stages of revision, the authors of the R15-sibs data reported that their earlier data release failed to correct properly for structure confounding. They released this corrected dataset as a replacement (*Robinson and Visscher, 2018*).

We also considered four different GWAS analyses of the UK Biobank data, using different subsets of individuals and different processing pipelines:

**UKB-GB**: ($n$ = 337 k) Linear regression controlling for 10 principal components of ancestry (unrelated British ancestry individuals only) (*Churchhouse and Neale, 2017*).

**UKB-Eur**: ($n$ = 459 k) All individuals of European ancestry, including relatives. Structure correction was performed using a Linear Mixed Model (LMM) approach, which controls for genetic stratification effects by modeling the genetic background as a random effect with covariance structure given by the kinship matrix. Mild amounts of environmental stratification are subsumed into this term, and therefore controlled implicitly (*Loh et al., 2017*).

**UKB-GB-NoPCs**: ($n$ = 337 k) Linear regression without any structure correction—with only genotype, age, sex and sequencing array as covariates (unrelated British ancestry individuals only) [newly calculated by us, see Materials and methods].

**UKB-sibs**: ($n$ = 35 k) Family-based sib-pair analysis [newly calculated by us, see Materials and methods].

To understand the extent to which these different datasets capture a shared signal, we treated each set of summary statistics as if it were derived from a GWAS of a different phenotype and estimated the genetic correlation between them using bivariate LD Score regression (*Bulik-Sullivan et al., 2015a*). We find that all of these studies show high pairwise genetic correlations , consistent with the view that all of them estimate a largely-similar genetic basis of height (*Table 2*).

**Table 2.** Pairwise genetic correlations between GWAS datasets.

Genetic correlation estimates (lower triangle) and their standard errors (upper triangle) between each of the height datasets, estimated using LD Score regression (**Bulik-Sullivan et al., 2015a**). All trait pairs show a strong genetic correlation, as expected for different studies of the same trait.

| | Giant | R15-sibs | UKB-Eur | UKB-GB-NoPCs | Ukb-gb | UKB-sibs |
|---|---|---|---|---|---|---|
| **GIANT** | | (0.04) | (0.01) | (0.01) | (0.01) | (0.05) |
| **R15-sibs** | 0.98 | | (0.04) | (0.04) | (0.05) | (0.08) |
| **UKB-Eur** | 1.03 | 0.87 | | (0.004) | (0.004) | (0.05) |
| **UKB-GB-NoPCs** | 1.01 | 0.82 | 1.00 | | (0.002) | (0.05) |
| **UKB-GB** | 1.03 | 0.89 | 1.02 | 1.00 | | (0.05) |
| **UKB-sibs** | 1.02 | 0.93 | 1.06 | 1.02 | 1.06 | |

DOI: https://doi.org/10.7554/eLife.39725.003

## Signal of selection across Eurasia

One well-studied signal of adaptation of height in Europe has been the observation that, among height-associated SNPs, the 'tall' alleles tend to be at higher frequencies in northern populations. Equivalently, the average polygenic scores of individuals in northern populations tend to be higher than individuals in southern populations. To evaluate this signal for each dataset, we independently ascertained the SNP with the smallest p-value within each of 1700 approximately independent LD blocks (**Berisa and Pickrell, 2016**; **Berg et al., 2017**) (subject to the constraint that MAF > 0.05 within the GBR 1000 Genomes population). We used these loci to calculate average polygenic scores for each of a set of European population samples taken from the 1000 Genomes and Human Origins panels (**Lazaridis et al., 2014**; **1000 Genomes Project Consortium et al., 2015**) (see Materials and methods for statistical details).

As expected, we find highly significant latitudinal gradients in both the GIANT and R15-sibs data. However, this signal does not replicate in any of the four UK Biobank datasets (**Figure 1**, top row), or in the R15-sibs-updated dataset (**Figure 1—figure supplement 1**).

We also tested whether the polygenic scores are over-dispersed compared to a neutral model, without requiring any relationship with latitude (the $Q_X$ test from **Berg and Coop, 2014**). Here we find a similar pattern: we strongly reject neutrality using both the GIANT and R15-sibs datasets, but see little evidence against neutrality among the UK Biobank datasets, or the R15-sibs-updated dataset (**Figure 1**). The sole exception is for the UKB-GB dataset, though the rejection of neutrality in this dataset is marginal compared to that observed with GIANT and R15-sibs, and it does not align with latitude.

While most studies have focused on a latitudinal cline in Europe, a preprint by **Berg et al. (2017)** also recently reported a cline of polygenic scores decreasing from west to east across all of Eurasia. Extending this analysis across all six datasets, we observe similarly inconsistent signals (**Figure 1**, bottom row). Only the GIANT dataset shows the clear longitudinal signal reported by **Berg et al. (2017)**, though the R15-sibs dataset is again strongly over-dispersed in general, and retains some of the longitudinal signal. Interestingly, we find a weakly significant relationship between longitude and polygenic score in the UKB-Eur dataset (though not in the other UKB datasets), suggesting there may be systematic differences between the results based on British-only and pan-European samples, even when state of the art corrections for population structure are applied.

We also experimented with a larger number of SNPs using a procedure similar to **Robinson et al. (2015)** (Appendix 1). We found that in some cases this led to significant values of $Q_X$ when using UKB-GB effect sizes to ascertain SNPs. However, this signal was sensitive to the particular method of ascertainment, and seems to be diffuse (i.e. spread out across all axes of population structure, **Appendix 1–figure 6**), with part of the signal coming from closely linked SNPs. Thus we conclude that this signal is not robust and may, at least partially, arise from a violation of the assumption of independence among SNPs that underlies our neutral model. We also tested different frequency, effect size and probability-of-association cutoffs to determine which SNPs we included in the computation of the scores, but found none of these cutoffs affected the discrepancy observed between the GIANT and UKB-GB datasets (Appendix 2).

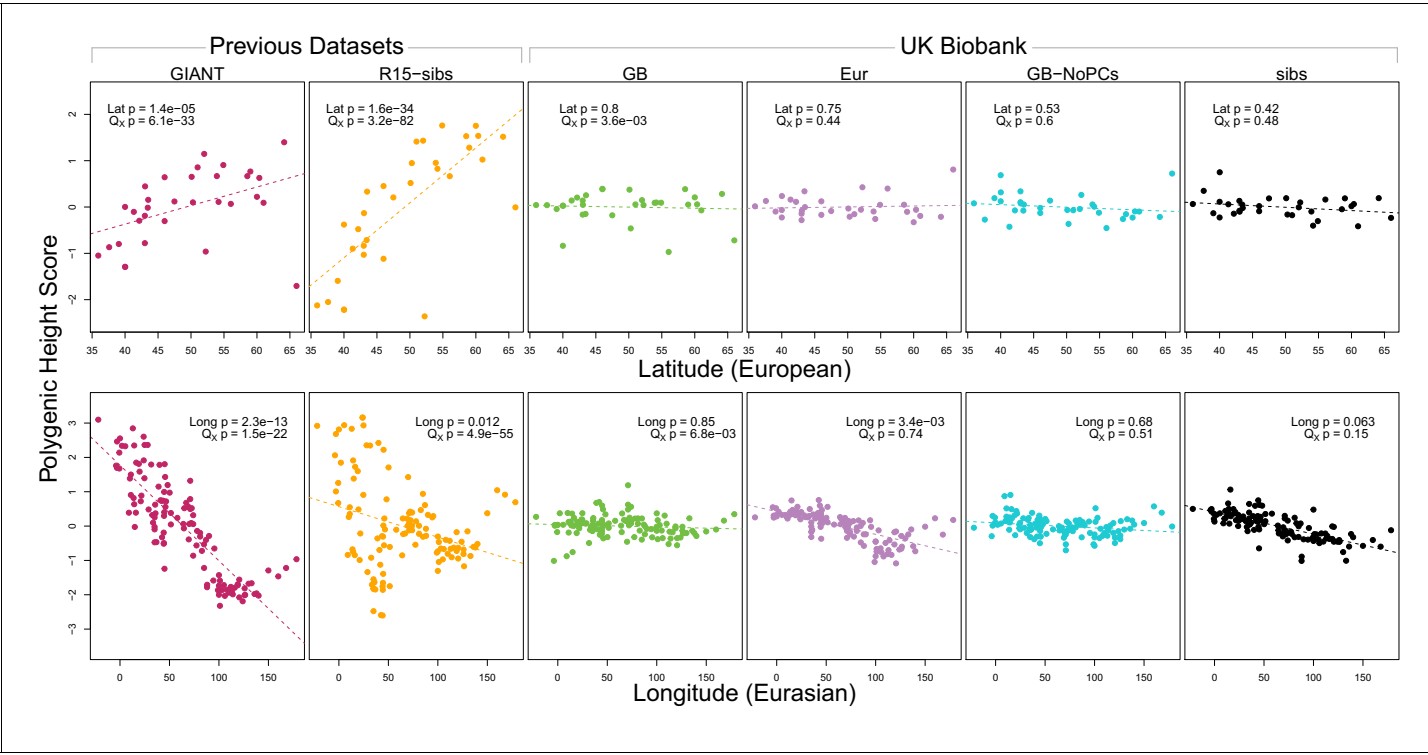

**Figure 1.** Polygenic scores across Eurasian populations, for different GWAS datasets. The top row shows European populations from the combined 1000 Genomes plus Human Origins panel, plotted against latitude, while the bottom row shows all Eurasian populations from the same combined dataset, plotted against longitude.

DOI: https://doi.org/10.7554/eLife.39725.004

The following figure supplement is available for figure 1:

**Figure supplement 1.** The R15-sibs-updated dataset shows no significant latitudinal or longitudinal signal.

DOI: https://doi.org/10.7554/eLife.39725.005

## SDS signal of selection in Britain

We next evaluated the Singleton Density Score (SDS) signal of selection for increased height in the British population, previously reported by *Field et al. (2016)*. SDS estimates recent changes in allele frequencies at each SNP within a population by comparing the distances to the nearest singleton variants linked to each of the focal SNP's alleles. (*Field et al., 2016*) applied SDS calculated across the UK10K sample (*UK10K Consortium, 2015*) to investigate allele frequency changes in the ancestors of modern British individuals. SDS can be polarized according to the sign of a GWAS effect at each SNP–this is denoted trait-SDS, or tSDS. Here, tSDS > 0 indicates that a height-increasing allele has risen in frequency in the recent past; tSDS < 0 correspondingly indicates a decrease in frequency of the height-increasing allele. A systematic pattern of tSDS > 0 is consistent with directional selection for increased height.

Using both GIANT and R15-sibs, *Field et al. (2016)* found a genome-wide pattern of positive tSDS, indicating that on average, height-increasing alleles have increased in frequency in the last ~75 generations. tSDS also showed a steady increase with the significance of a SNP's association with height. We replicate these trends in *Figure 2A,B*.

This tSDS trend is greatly attenuated in all four GWAS versions performed on the UK Biobank sample (*Figure 2C–F*), as well as the R15-sibs-updated dataset (*Figure 2—figure supplement 1*). The correlation between UKB-GB GWAS p-value and tSDS is weak (Spearman $\rho = 0.009$, block-jack-knife $p = 0.04$). This correlation is stronger for the UKB-Eur GWAS ($\rho = 0.018$, $p = 5 \times 10^{-7}$). Since the UKB-Eur GWAS is not limited to British individuals—but instead includes all European ancestry individuals—this might suggest that residual European population structure continues to confound

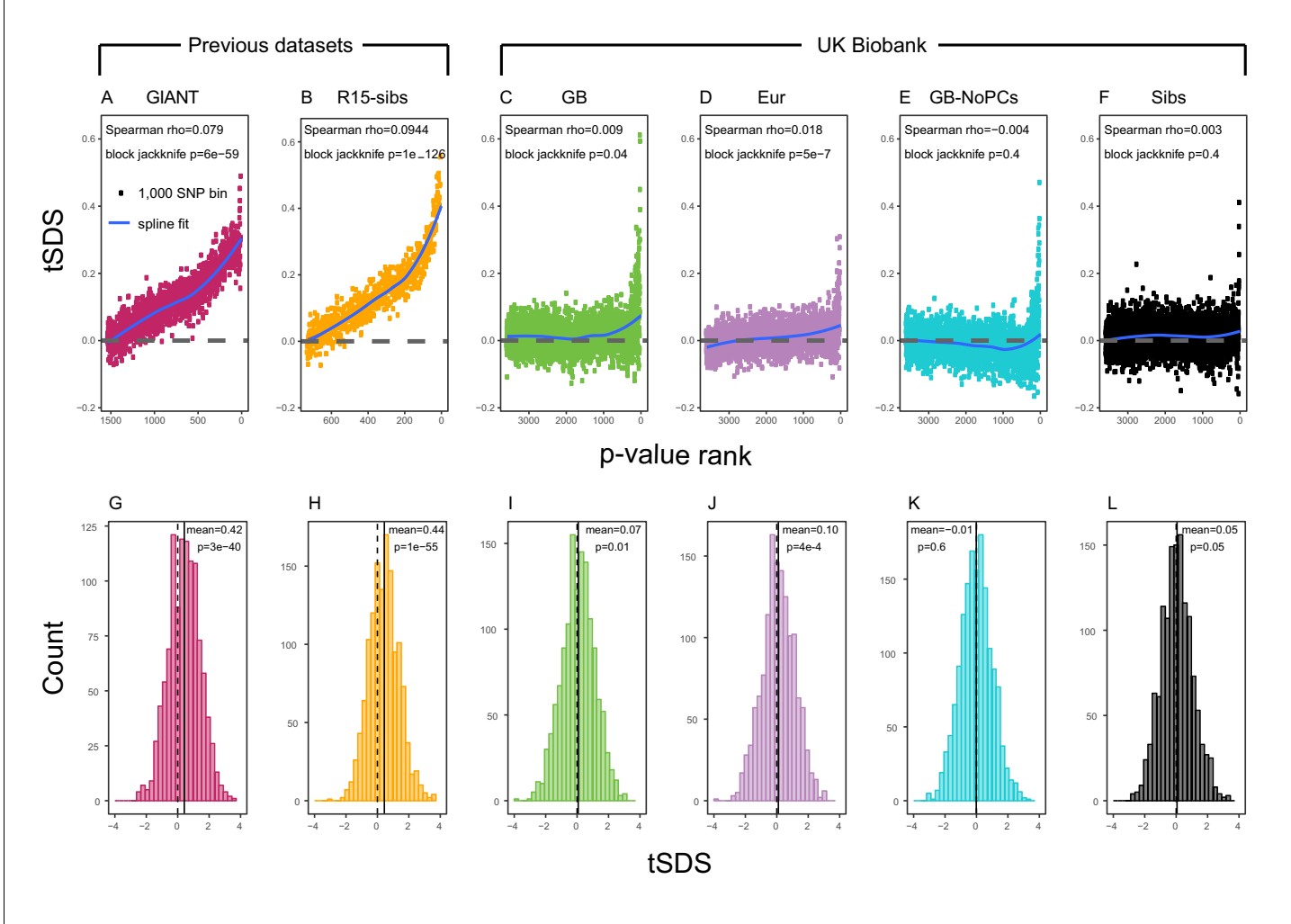

**Figure 2.** SDS signals for recent selection, assessed using different height GWAS. (**A–F**) Each point shows the average tSDS (SDS polarized to height-increasing allele) of 1000 consecutive SNPs in the ordered list of GWAS p-values. Positive values of tSDS are taken as evidence for selection for increased height, and a global monotonic increase—as seen in panels A and B—suggests highly polygenic selection. (**G–L**) tSDS distribution for the most significant SNPs in each GWAS, thinned according to LD to represent approximately independent signals. Dashed vertical lines show tSDS = 0, as expected under the neutral null; solid vertical lines show mean tSDS. A significantly positive mean value of tSDS suggests selection for increased height.

DOI: https://doi.org/10.7554/eLife.39725.006

The following figure supplement is available for figure 2:

**Figure supplement 1.** Previously reported SDS selection signals are also absent from the R15-sibs-updated dataset.
DOI: https://doi.org/10.7554/eLife.39725.007

UKB-Eur effect estimates, despite the use of LMM correction for structure, similar to the longitudinal signal detected above for this same dataset (*Figure 1*).

We wondered whether the main reason for the weakened trend in UKB-GB is an overly conservative PC-correction. This could occur if the genetic contribution to height is highly correlated with population structure axes. If this were the case, we would expect the correlation between GWAS p-value and tSDS to still be observed in a UKB GWAS without population structure correction (namely, in UKB-GB-NoPCs). However, we see no evidence for this correlation (block jackknife p = 0.6). Taken together with the UKB-GB-NoPCs polygenic score analysis (*Figure 1*), the lack of signal in UKB-GB-NoPCs suggests that the main reason that UKB is less confounded by population structure than GIANT is the relatively-homogeneous ancestry of the UKB British sample—rather than differences in GWAS correction procedures.

Lastly, we examined tSDS at the most significant height-associated SNPs of each UKB GWAS (as before, ascertained in approximately-independent LD blocks). Significant SNPs show a positive average tSDS (*Figure 2I,K,L*; t-test p < 0.05)—with the exception of the UKB-GB-NoPCs GWAS (*Figure 2J*) in which the average tSDS is not significantly different from zero (t-test p = 0.6).

## Relationship between GWAS estimates and European population structure

We have now shown that signals of polygenic adaptation of height are greatly reduced in the UKB data relative to the GIANT and R15-sibs datasets. To better understand the differences among the datasets, we ascertained 1,652 approximately-independent lead SNPs based on the GIANT p-values to form the basis of comparison between the GIANT and UKB-GB datasets.

*Figure 3A* shows the effect sizes of ancestral alleles, as estimated using GIANT (x-axis) and UKB-GB (y-axis). The two datasets are highly correlated ($r^2 = 0.78$, $p < 2.2 \times 10^{-16}$), consistent with the strong genetic correlation estimated in *Table 2*. The fact that the slope is <1 probably reflects, at least in part, the standard winner's curse effect for SNPs ascertained in one study and replicated in another.

Importantly however, we also see clear evidence that the *differences* between the GIANT and UKB-GB effect sizes are correlated with European population structure (*Figure 3B*). Specifically, for each SNP we plotted the difference in effect size between GIANT and UKB-GB against the difference in allele frequency between northern and southern European samples (specifically, between the British (GBR) and Tuscan (TSI) subsets of 1000 Genomes) . These differences have a significant correlation ($r^2 = 0.06$, $p < 2.2 \times 10^{-16}$), indicating that alleles that are more frequent in GBR, compared to TSI, tend to have more positive effect sizes in GIANT than in UKB-GB, and vice versa. We also observed a similar signal for frequency differences between TSI and the Han Chinese in Beijing (CHB, *Figure 3—figure supplement 1*), suggesting that longitudinal patterns observed by *Berg et al. (2017)* were also likely driven by incompletely controlled stratification in GIANT.

Similar patterns are present in a comparison of the R15-sibs and UKB-GB datasets when ascertaining from R15-sibs p-values (*Figure 3*, panels C and D; 1,642 SNPs). Here, the correlation between effect size estimates is much lower ($r^2 = 0.14$, $p < 2.2 \times 10^{-16}$), likely due to the much smaller sample size of R15-sibs, and therefore elevated winner's curse effects. However, the correlation between the effect-size difference and the GBR-TSI allele frequency difference remains ($r^2 = 0.07$, $p < 2.2 \times 10^{-16}$). In contrast, when SNPs are ascertained on the basis of their UKB-GB p-value, these patterns are considerably weaker in both the GIANT and R15-sibs datasets (*Figure 3—figure supplement 2*).

Finally, an unexpected feature of the R15-sibs dataset can be seen in *Figure 3D*: there is a strong skew for the ancestral allele to be associated with increased height (1,201 out of the 1,642 SNPs ascertained with R15-sibs p-values have positive effect sizes in R15-sibs). This pattern is not present in the R15-sibs-updated dataset (851 out of 1,699 SNPs with positive effects), or any other dataset we analyzed, suggesting that it is likely a result of the failure to control for population structure.

Together, these observations suggest that while all of the datasets primarily capture real signals of association with height, both the GIANT and R15-sibs effect size estimates suffer from confounding along major axes of variation in Europe and Eurasia. This could drive false positive signals in geographic-based analyses of polygenic adaptation. Furthermore, since SDS measured in Britain correlates with north-south frequency differences (*Field et al., 2016*), this could also drive false positives for SDS.

To explore this further, we next turn to an analysis of the datasets based on LD Score regression.

## LD Score regression signal

Another line of evidence from *Field et al. (2016)* came from LD Score regression (*Bulik-Sullivan et al., 2015a*; *Bulik-Sullivan et al., 2015b*). LD Score regression applies the principle that, under a polygenic model, SNPs in regions of stronger LD (quantified by a SNP's 'LD Score') should tag more causal variants and therefore have larger squared effect size estimates. Similarly, if two traits share a genetic basis, then the correlation between estimated effect sizes of these traits should increase with LD Score. Meanwhile, confounders such as population structure are argued to affect

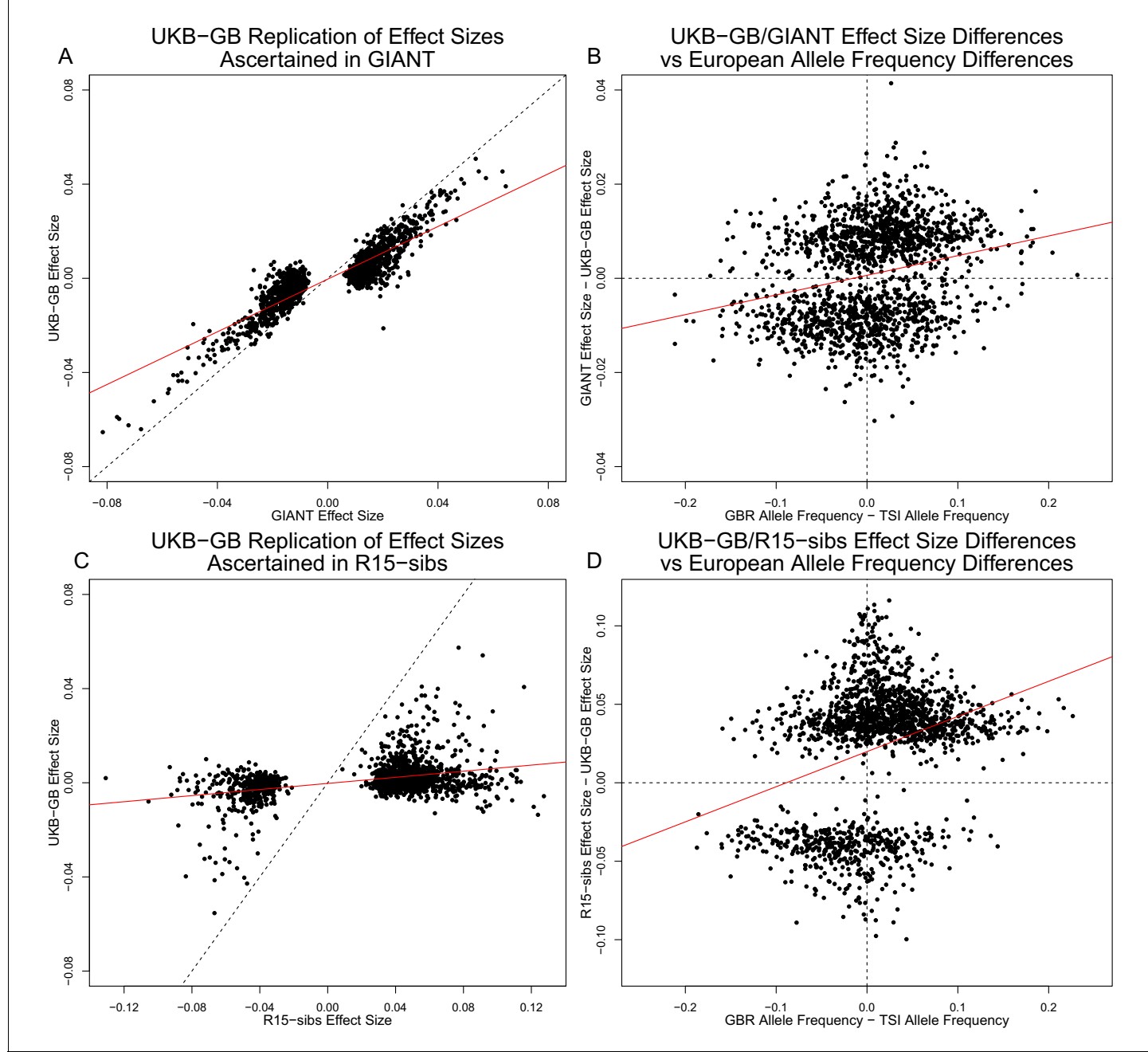

**Figure 3.** Effect size estimates and population structure. Top Row: SNPs ascertained using GIANT compared with UKB-GB. (**A**) The x- and y-axes show the estimated effect sizes of SNPs in GIANT and in UKB-GB. Note that the signals are highly correlated overall, indicating that these partially capture a shared signal (presumably true effects of these SNPs on height). (**B**) The x-axis shows the difference in ancestral allele frequency for each SNP between 1000 Genomes GBR and TSI; the y-axis shows the difference in effect size as estimated by GIANT and UKB-GB. These two variables are significantly correlated, indicating that a component of the difference between GIANT and UKB-GB is related to the major axis of population structure across Europe. Bottom Row: SNPs ascertained using R15-sibs compared with UKB-GB. (**C**) The same plot as panel (**A**), but ascertaining with and plotting R15-sibs effect sizes rather than GIANT. Here, the correlation between effect size estimates of the two studies is reduced relative to panel A–likely due to the lower power of the R15-sibs study compared to GIANT (**D**). Similarly, the same as (**B**), but with the R15-sibs ascertainment and effect sizes.

DOI: https://doi.org/10.7554/eLife.39725.008

The following figure supplements are available for figure 3:

**Figure supplement 1.** Similar patterns are seen for a TSI-CHB frequency contrast, suggesting the longitudinal patterns seen with GIANT data were also a result of stratification.

DOI: https://doi.org/10.7554/eLife.39725.009

*Figure 3 continued on next page*

*Figure 3 continued*

**Figure supplement 2.** The strength of the correlation between effect size difference and frequency difference is much reduced when ascertained using UKB-GB, suggesting a significant effect of ascertainment bias.
DOI: https://doi.org/10.7554/eLife.39725.010

SNPs of different LD Score equally, and therefore affect the intercept but not the slope of a linear regression to LD Score (we return to this point below; *Bulik-Sullivan et al., 2015b*).

While LD Score regression is commonly used to estimate the genetic covariance between pairs of phenotypes (*Bulik-Sullivan et al., 2015a*), *Field et al. (2016)* used it to test for a relationship between height and SDS. SDS is similar to GWAS effect estimates in that the expected change in frequency of an allele depends on both direct selection it experiences due to its own fitness effect as well as correlated selection due to the effects of those in linkage disequilibrium with it. *Field et al. (2016)* predicted that the covariance between estimated marginal height effect and SDS should increase with LD Score—and found this to be the case using both GIANT and R15-sibs. This provided further evidence for polygenic adaptation for increased stature in Britain.

Here, we revisit *Field et al. (2016)*'s observations (*Figure 4A,B*). Both GIANT and R15-sibs exhibit a highly significant LD Score regression slope (scaled GIANT slope = 0.17, $p = 5 \times 10^{-9}$; scaled R15-sibs slope = 0.46, $p = 7 \times 10^{-17}$), as well as a highly significant intercept (GIANT intercept = 0.093, $p = 4 \times 10^{-71}$; R15-sibs intercept = 0.119, $p = 2 \times 10^{-87}$). These large intercepts suggest that both GWAS suffer from stratification along an axis of population structure that is correlated with SDS in the British population. In contrast, in LD Score regression with the UKB-GB GWAS, the intercept is not significant ($p = 0.10$), suggesting that UKB-GB is not strongly stratified (or at least, not along an axis that correlates with SDS). The slope is ~1/3 as large as in GIANT, though still modestly significant ($p = 1.2 \times 10^{-2}$, *Figure 4C*). There is no significant slope ($p = 0.389$) or intercept ($p = 0.405$) in R15-sibs-updated (*Figure 4—figure supplement 1B*), and analyses of other UKB datasets give similar results to those for UKB-GB (*Figure 4—figure supplement 2*).

## LD Score regression of population frequency differences

We next extended Field et al.'s LD Score rationale from SDS to test whether SNP effects on height affected allele frequency differentiation between northern and southern Europe. We used 1000 Genomes British (GBR) and Tuscan (TSI) samples as proxies for northern and southern ancestry respectively. To control for the correlation between allele frequencies and LD Score, we normalized the frequency differences to have variance 1 within 1% average minor allele frequency bins. For shorthand, we refer to this measure as [GBR-TSI]. Under a model of selection driving allele frequency differences, we would expect the covariance of [GBR-TSI] and effect sizes to increase with LD Score. To test this, we regressed the product [GBR-TSI] × effect size (estimated in previous and UKB GWAS) against LD Score.

In contrast to SDS, we find that none of the GWAS datasets show a strongly positive slope (*Figure 4D–F*): the slope is approximately zero in GIANT, weakly positive in R15-sibs ($p = 0.002$), and weakly negative in UKB-GB ($p = 0.09$) Results were similar for the other UKB datasets (*Figure 4—figure supplement 3*), and for R15-sibs-updated (*Figure 4—figure supplement 1A*). We see extremely strong evidence for positive intercepts in GIANT (p = $4 \times 10^{-80}$) and R15-sibs (p = $9 \times 10^{-161}$), but not in UKB-GB ($p = 0.05$), R15-sibs-updated ($p = 0.848$) or the other UKB datasets. The large intercepts in GIANT and R15-sibs are consistent with stratification affecting both of these GWAS, as the North-South allele frequency difference is systematically correlated with the effect sizes in these GWAS independently of LD Score (see the Materials and methods for a more technical discussion). However, the relative lack of slope in these analyses suggests that the LD Score signal for SDS must be driven by a component of frequency change that is largely uncorrelated with the [GBR-TSI] axis of variation.

## Population structure confounds LD Score regression slope

The original LD Score regression paper noted that in the presence of linked selection, allele frequency differentiation might plausibly increase with LD Score. However, they concluded that this effect was negligible in the examples they considered (*Bulik-Sullivan et al., 2015b*), and subsequent

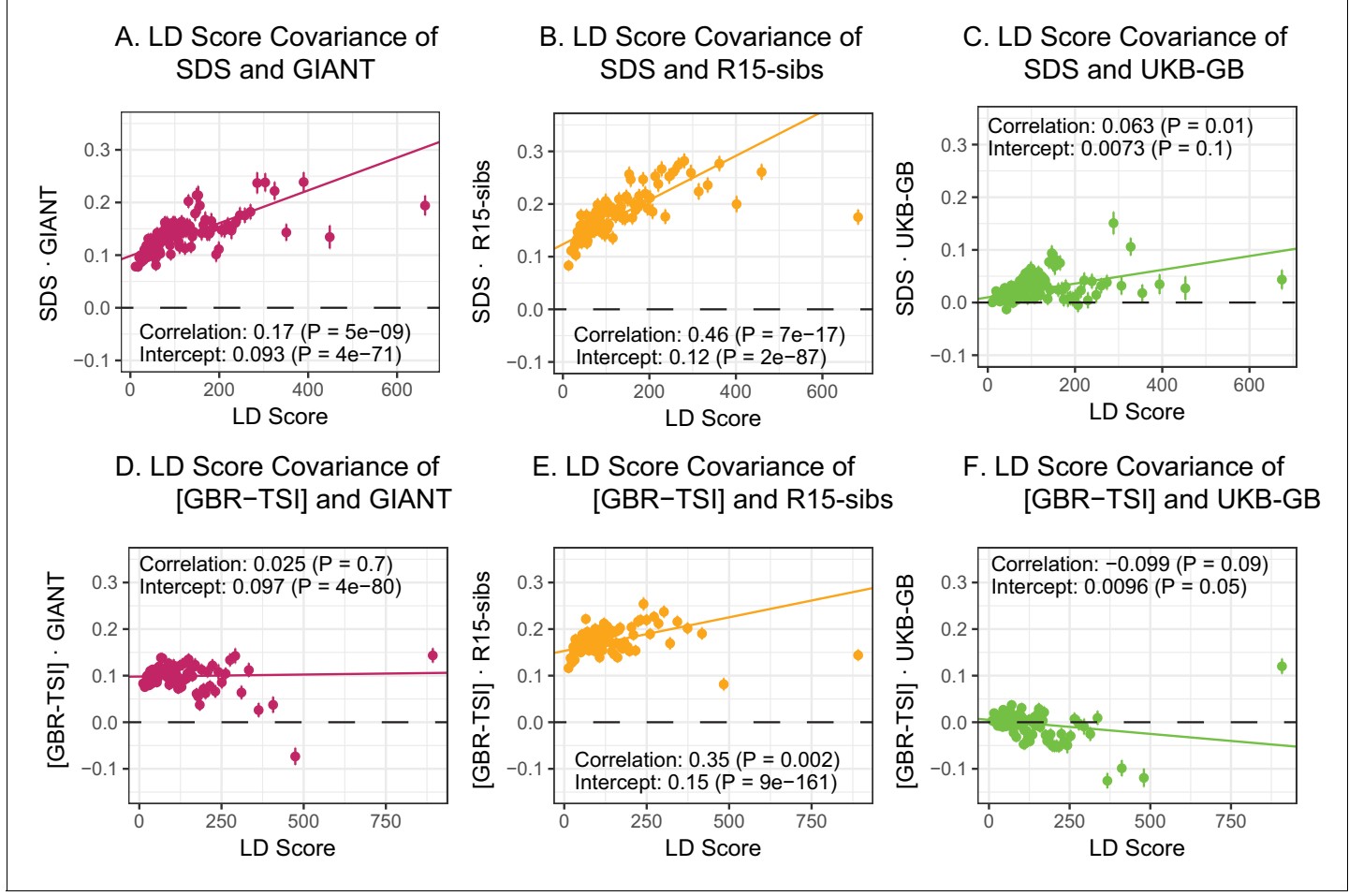

**Figure 4.** LD Score regression analyses. (**A**), (**B**), and (**C**) LD Score covariance analysis of SDS with GIANT, R15-sibs, and UKB-GB, respectively. The x-axis of each plot shows LD Score, and the y-axis shows the average value of the product of effect size on height and SDS, for all SNPs in a bin. Genetic correlation estimates are a function of slope, reference LD Scores, and the sample size *Bulik-Sullivan et al. (2015a)*. Both the slope and intercept are substantially attenuated in UKB-GB. (**D**), (**E**) and (**F**) Genetic covariance between GBR-TSI frequency differences vs. GIANT, R15-sibs, and UKB-GB. GIANT and R15-sibs show highly significant nonzero intercepts, consistent with a signal of population structure in both datasets, while UKB-GB does not. In addition, R15-sibs shows a significant slope with LD Score.
DOI: https://doi.org/10.7554/eLife.39725.011

The following figure supplements are available for figure 4:

**Figure supplement 1.** The R15-sibs-updated dataset shows no evidence of LD Score regression slope with [GBR-TSI] or with SDS.
DOI: https://doi.org/10.7554/eLife.39725.012

**Figure supplement 2.** UK Biobank datasets show little evidence of bivariate LD Score regression slope when analyzed together with SDS.
DOI: https://doi.org/10.7554/eLife.39725.013

**Figure supplement 3.** Similarly, no dataset has a significant positive slope for [GBR-TSI].
DOI: https://doi.org/10.7554/eLife.39725.014

applications of the LD Score regression approach have generally assumed that the two are independent. We find that in bivariate LD Score analyses of SDS, both the intercept and the slope differ significantly from zero for precisely the same GWAS datasets that show evidence of stratification (GIANT and R15-sibs), while both the slope and intercept are much reduced in all of the UK Biobank datasets. This suggests that the LD Score regression slope may not be as robust to stratification as hoped, prompting us to revisit the assumption of independence.

We find that the squared allele frequency difference [GBR-TSI]$^2$ is significantly correlated with LD Score ($p = 2.5 \times 10^{-5}$, *Figure 5A*), as are squared allele frequency contrasts for much lower levels of differentiation (i.e. between self-identified 'Irish' and 'White British' individuals in the UK Biobank

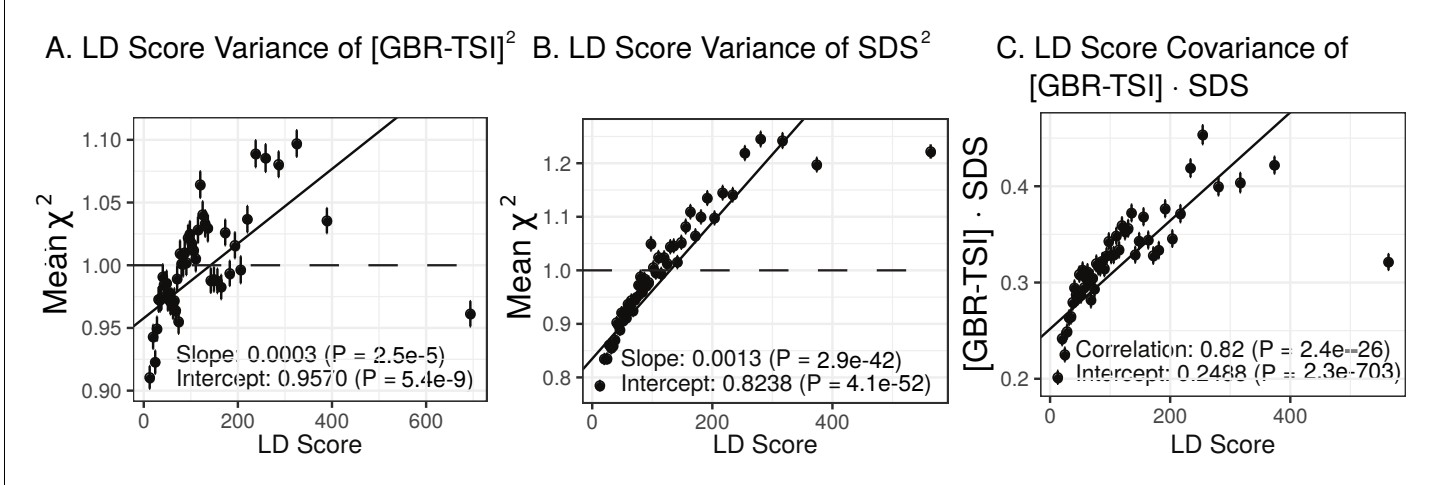

**Figure 5.** Population allele frequencies show genetic correlation with European height GWAS. (A), (B) and (C) Magnitude of squared GBR-TSI allele frequency differences, squared SDS effect sizes, and the product of allele frequency and SDS increase with LD Score. Both SDS and GBR-TSI frequency difference are standardized and normalized within 1% minor allele frequency bins..

DOI: https://doi.org/10.7554/eLife.39725.015

The following figure supplement is available for figure 5:

**Figure supplement 1.** LD Score regression results for the difference in allele frequency between individuals who identified as Irish and those who identified as White British in the UK Biobank (related individuals removed).

DOI: https://doi.org/10.7554/eLife.39725.016

($p = 2.5 \times 10^{-7}$, *Figure 5—figure supplement 1*), and SDS$^2$ ($p = 2.9 \times 10^{-42}$, *Figure 5B*). Strikingly, squared measures of more recent allele frequency change (i.e. SDS$^2$ and the squared Irish vs. White British contrast) are much more tightly correlated with LD Score than that of more diverged populations [GBR-TSI]$^2$, suggesting that the LD Score regression slope may be equally vulnerable to stratification involving closely related populations than for those that are more distantly related. Finally, the product [GBR-TSI] $\times$ SDS is also correlated with LD Score (*Figure 5C*), demonstrating that the general signal of greater allele frequency change in regions of stronger LD is also shared between [GBR-TSI] and SDS.

## Background selection and LD Score

As noted above, the correlation we observe between allele frequency differentiation and LD Score could be generated by the genome-wide effects of linked selection. While a range of different modes of linked selection likely act in humans, one of the simplest is background selection (*Charlesworth et al., 1993*; *Charlesworth, 1998*; *McVicker et al., 2009*). Background selection (BGS) on neutral polymorphisms results from the purging of linked, strongly deleterious alleles. Because any neutral allele that is in strong LD with a deleterious mutation will also be purged from the population, the primary effect of BGS is a reduction in the number of chromosomes that contribute descendants in the next generation. The impact of BGS can therefore be approximately thought of as increasing the rate of genetic drift in genomic regions of strong LD relative to regions of weak LD. Therefore, SNPs with larger LD Scores will experience stronger BGS and a higher rate of genetic drift, and this effect could generate a positive relationship between LD Scores and allele frequency differentiation.

In Appendix 3, we derive a simple model for the effect of BGS on the relationship between allele frequency divergence and LD Scores. Empirically, we find that LD Scores are positively correlated with the strength of background selection (*Appendix 3—figure 1*) (*McVicker et al., 2009*), and that our simple model of background selection is capable of explaining much of the relationship between LD Score and allele frequency divergence that we observe in *Figure 5* (*Appendix 3—figures 1–3*). Further, in the presence of BGS, bivariate regression of a stratified GWAS together with a

measure of allele frequency differentiation can result in a positive slope, provided that the axis of stratification is correlated with the chosen measure of allele frequency divergence (Appendix 3).

## Summary of LD Score regression results

What conclusions should we draw from our LD Score regression analyses? The significant positive intercepts observed for LD Score regression of both GIANT and R15-sibs with [GBR-TSI] suggest that both datasets suffer from confounding due to stratification along a north-to-south axis within Europe. These observations are consistent with the evidence presented in *Figure 3*. A positive slope in such analyses was previously interpreted as evidence of positive selection either on height or a close genetic correlate (and presumed to be robust to stratification). However, BGS can, and empirically does, violate the assumptions of LD Score regression in a way that may generate a positive slope. We therefore interpret the positive slopes observed for the LD Score regression signals for GIANT and R15-sibs with SDS as likely resulting from a combination of stratification and BGS. A similar conclusion applies to the positive slope observed for R15-sibs × [GBR-TSI]. It is unclear why stratification plus BGS should have elevated the slope for GIANT × SDS, but not for GIANT × [GBR-TSI]. This may suggest that the apparent SDS selection signal found in GIANT may be driven by an axis of variation that is not strongly correlated with [GBR-TSI]. We view this as an area worthy of further exploration.

## Discussion

To summarize the key observations, we have reported the following:

- Multiple analyses based on GIANT and R15-sibs indicate strong signals of selection on height.
- However, the same signals of selection are absent or greatly attenuated in UK Biobank data. In some, but not all, analyses of frequency differentiation and SDS we still detect weakly significant signals of polygenic adaptation (*Figures 1* and *2*).
- The GIANT height GWAS is overall highly correlated with UKB-GB, but differs specifically by having an additional correlation with the main gradient of allele frequency variation across Europe, as modeled by frequency differences between GBR and TSI (*Figure 3*). LD Score analysis of [GBR-TSI] × GIANT effect-size also suggests that GIANT is stratified along this axis (positive intercept in *Figure 4D*).
- Selection signals in the R15-sibs data are consistent with, and in some cases even stronger, than the corresponding signals in GIANT, but are inconsistent with analyses using UK Biobank data. While correctly implemented sib-based studies are designed to be impervious to population structure, [GBR-TSI] × R15-sibs effect size also shows a highly positive intercept in the LD Score analysis presented in *Figure 4E*. As discussed below, the R15-sibs authors have recently identified a bug in the pipeline that generated this dataset. Analysis of corrected summary statistics does not show such a signal (*Figure 4—figure supplement 1*).
- LD Score analyses show a much stronger relationship between SDS and GIANT or R15-sibs than between SDS and UKB-GB. LD Score regression is generally considered to be robust to population structure (but see the discussion in *Bulik-Sullivan et al., 2015b*). However, the intensity of background selection increases with LD Score (*Figure 5*, Appendix 3), and this has likely inflated the LD Score-based signal of selection in GIANT and R15-sibs.

In principle, it is possible that height in the UKB is confounded in a way that suppresses the signal of height adaptation. Instead, multiple lines of evidence strongly suggest that population-structure confounding in GIANT and R15-sibs is the main driver of the discrepancy with UKB-based analyses.

The sib design used by Robinson et al offered a strong independent replication of the polygenic adaptation signal, which should have been impervious to population structure concerns. However, our analyses highlight multiple signs of stratification in this study. Robinson et al have now confirmed (as of November 2018) that effect sizes they released from their 2015 study were strongly affected by population stratification due to a bug. Furthermore, they have now stated that the effect sizes that they released publicly in 2016 were not the effect sizes used in their 2015 paper. As part of our own investigation, we have independently confirmed that sib-studies conducted using PLINK v1.90b5 are robust to environmental stratification (*Appendix 4–figure 1*). Our analyses using the newly rerun effect sizes released by *Robinson and Visscher, 2018* show no consistent signal of

selection (*Figure 1—figure supplement 1*, *Figure 2—figure supplement 1*. and *Figure 4—figure supplement 1*), in line with our UKB-based analyses.

GIANT was conducted as a collaboration among a large number of research groups that provided summary statistics to the overall consortium. While the overall value of this pioneering dataset is not in question, it would not be surprising in retrospect if this GWAS were impacted by residual stratification along major axes of population structure.

Lastly, we must conclude that the strong signal of LD Score genetic covariance between SDS and both GIANT and R15-sibs is largely spurious. This would imply that the LD Score regression slope is not robust to population structure confounding. Specifically, we demonstrated that background selection—through its correlation with LD Score—can potentially generate a spurious LD Score regression slope.

Taken together, these observations lead us to conclude that what once appeared an ironclad example of population genetic evidence for polygenic adaptation now lacks any strong support. That said, there is still strong evidence that typical GWAS, including GIANT, do capture genuine signals of genotype-phenotype associations. For example, GWAS datasets regularly show strong functional enrichments of heritability within active chromatin from trait-relevant tissues (*Finucane et al., 2015*), and the observation that top SNPs identified in GIANT tend to replicate in UKB-GB (*Figure 3A*), together with the high genetic correlations among all of the datasets (*Table 2*), suggest that the vast majority of the signal captured by GIANT is real.

Nonetheless, we have shown that GIANT effect-size estimates contain a component arising from stratification along a major axis of European population structure (*Figure 3B*), and one would like to know the extent to which the conclusions from other analyses of GIANT, or other GWAS, may be affected. A complete investigation of this is beyond the scope of this study, and will depend on the nature of the analyses performed. The problem is likely most acute for the analysis of polygenic scores in samples drawn from heterogeneous ancestry. This is because while the bias in detection and effect sizes at any individual locus is small, the systematic nature of biases across many loci compound to significant errors at the level of polygenic scores. This error substantially inflates the proportion of the variance in polygenic scores that is among populations. Individual level prediction efforts therefore suffer dramatically from stratification bias, as even small differences in ancestry will be inadvertently translated into large differences in predicted phenotype (*Kerminen et al., 2018*). This seems likely to remain a difficult complication even within datasets such as the UK Biobank, though we suspect that meta-analyses such as GIANT, which collate summary statistics from many sources, may be particularly sensitive to structure confounding.

These issues are apparent even within our UK Biobank results, where we see marked differences between results based on UKB-GB and UKB-Eur (*Figure 2C,I* vs. D, J and *Figure 4—figure supplement 3*). The study subjects in the two datasets were largely overlapping, and both were computed using widely-accepted structure-correction methods, suggesting that in the more demanding setting of broad European ancestry variation, the linear mixed model approach did not provide complete protection against stratification. This highlights a need for renewed exploration of the robustness of these methods, especially in the context of polygenic prediction.

The study of polygenic scores across ancestry and environmental gradients offers a range of promises and pitfalls (*Berg and Coop, 2014*; *Novembre and Barton, 2018*). Looking forward, we recommend that studies of polygenic adaptation should focus on datasets that minimize population structure (such as subsets of UKB), and where the investigators have access to full genotype data, including family data, so that they can explore sensitivity to different datasets and analysis pipelines.

## Materials and methods

### Newly calculated GWAS

*Figure 1* and *Figure 2* display analyses based on six different GWAS. Two of these GWAS were newly calculated by us using UK Biobank data. Below, we describe the specifics of these two GWAS.

#### UKB-GB-NoPCs

To preform this GWAS, we used the following *plink v. 2.0* (*Purcell et al., 2007*) with command line as follows:

- plink2 –memory 64G –threads 16 –linear
- –bpfile ⟨UKB imputed SNPs bp file⟩
- –keep ⟨id list of individuals self-identified as 'White British'⟩
- –out ⟨output file⟩
- –pheno ⟨standing height phenotype file (UKB phenotype 50.0.0)⟩
- —covar ⟨covariates file⟩

The covariates file included only the sex, age and sequencing array for each individual id. We filtered all A↔G or C↔T SNPs–to prevent the possibility of strand errors. Finally, we excluded SNPs for which SDS was not calculated in *Field et al. (2016)*.

## UKB-sibs
We used the estimated kinship coefficient ($\phi$) and the proportion of SNPs for which the individuals share no allele (IBS0) provided by the UK Biobank, to call siblings as pairs with

$$\frac{1}{2^{5/2}} < \phi < \frac{1}{2^{3/2}}$$

and IBS0 >0.0012—following the conditions used by *Bycroft et al. (2018)*. We further filtered sibling pairs such that both individuals were 'White British', their reported sex matched their inferred sex, were not identified by the UK Biobank as 'outliers' based on heterozygosity and missing rate nor had an excessive number relatives in the data, and had height measurements. We standardized height values for each sex based on its mean and standard deviation (SD) values in the sample of 336,810 unrelated British ancestry individuals: mean 175.9 cm and SD 6.7 cm for males, and mean 162.7 cm and SD 6.2 cm for females. We also removed pairs if one of the siblings was more than 5 SD away from the mean. After applying all filters, 19,268 sibling pairs remained, equaling 35,524 individuals in 17,275 families. We performed an association analysis on 10,879,183 biallelic SNPs included in UKB-GB (converting dosages from imputation to genotype calls using no hard calling threshold), using *plink* v. 1.9 (*Purcell et al., 2007*) QFAM procedure with the following command:

- plink –bfile ⟨UKB hard-called SNPs file⟩
- -out ⟨output file⟩
- -qfam mperm = 100000

The family relationships, as well as the phenotypic values, were encoded in *plink* FAM files.

## GBR-TSI allele frequency differences
Individuals from the GBR and TSI populations from 1000G Phase 3 (N = 189) (*1000 Genomes Project Consortium et al., 2015*) were assigned binary phenotype labels and a $\chi^2$ test was run using plink (*Purcell et al., 2007*) with a Hardy-Weinberg equilibrium cutoff of 1e-6 (`–hwe 1e-6`) and missing genotype rate of 0.05 (`–geno 0.05`), but otherwise with default parameters. Additionally, a firth adjusted logistic regression (*Firth, 1993*) was run and produced qualitatively similar results (data not shown).

## IRL-GBR allele frequency differences
Unrelated individuals, defined using estimates from KING (*Manichaikul et al., 2010*), who self-identified as White British or White Irish in the UK Biobank were compared with distinct phenotype labels. Logistic regression (*Hill et al., 2017*) was run on the genotyped SNP set using plink2 (*Chang et al., 2015*) with a Hardy-Weinberg equilibrium cutoff of 1e-6 (`–hwe 1e-6`) and missing genotype rate of 0.05 (`–geno 0.05`).

## Polygenic score analyses
### Population genetic datasets
1000 Genomes Phase 3 VCF files were downloaded from the 1000 Genomes website, and VCF files for the Human Origins dataset were downloaded from the 'Affymetrix Human Origins fully public dataset' link on the Reich lab website and subsequently imputed to full genomes using the Michigan imputation server (*Das et al., 2016*). Because the Human Origins panel includes some 1000

Genomes populations, individual IDs were compared between the two datasets, and any duplicates were removed from the Human Origins dataset. Individuals were then clustered into populations based on groupings provided by each data resource, and allele frequencies were calculated using VCFtools version 0.1.15.

## Neutrality tests

In *Figure 1*, we employ two separate tests to assess the evidence that the distribution of polygenic scores among populations is driven in part by adaptive divergence. Both are based on a simple null model introduced by *Berg and Coop (2014)*, which states that the distribution of polygenic scores under neutrality should be approximately multivariate normal. Here, we give a brief overview of the assumptions and calculations underlying the null model, before describing the two tests used in *Figure 1*. For a more complete treatment, see *Berg and Coop (2014)*.

Let $\vec{p}_\ell$ be the vector of population allele frequencies at SNP $\ell$, while $\alpha_\ell$ is the effect size for SNP $\ell \in \{1, ..., L\}$. Then, population level polygenic scores are given by

$$\vec{Z} = 2\sum_\ell \alpha_\ell \vec{p}_\ell. \tag{1}$$

Under neutrality, the distribution of polygenic scores among populations should be approximately

$$\vec{Z} \sim MVN\left(\mu\vec{1}, 2V_A\mathbf{F}\right) \tag{2}$$

where

$$\mu = 2\sum_\ell \alpha_\ell \epsilon_\ell \tag{3}$$

$$V_A = 2\sum_\ell \alpha_\ell^2 \epsilon_\ell(1 - \epsilon_\ell) \tag{4}$$

where $\epsilon_\ell$ is the mean of $\vec{p}_\ell$ across populations. The matrix $\mathbf{F}$ gives the population level co-ancestry among populations. Here, we calculate the matrix $\mathbf{F}$ directly from the same set of SNPs used to calculate polygenic scores, which is a conservative procedure. Concretely, let

$$\vec{x}_\ell = \frac{\vec{p}_\ell - \epsilon_\ell}{\sqrt{\epsilon_\ell(1 - \epsilon_\ell)}}. \tag{5}$$

Then, if $\mathbf{X}$ is a matrix with the $\vec{x}_\ell$ as columns, we have

$$\mathbf{F} = \frac{1}{L-1}\mathbf{X}\mathbf{X}^T. \tag{6}$$

Now, based on this null model, we perform two separate neutrality tests. One is a general over-dispersion test (i.e. the '$Q_X$ test' from *Berg and Coop, 2014*), for which the test statistic is

$$Q_X = \frac{(\vec{Z} - \mu)^T \mathbf{F}^{-1}(\vec{Z} - \mu)}{2V_A}. \tag{7}$$

For $M$ populations, this statistic is expected to have a $\chi^2_{M-1}$ distribution under the multivariate normal null model (*Equation 2*). An unusually large value of $Q_X$ indicates that the neutral null model is a poor fit, and is therefore taken as evidence in favor of selection.

We also apply a second, more specific test, to test for evidence of a correlation with a specific geographic axis that is unusually strong compared to the neutral expectation. For any vector $\vec{Y}$, if $\vec{Z}$ has a multivariate normal distribution given by *Equation 2*, then

$$\vec{Y}^T\vec{Z} \sim N\left(\mu\vec{Y}^T\vec{1}, 2V_A\vec{Y}^T\mathbf{F}\vec{Y}\right) \tag{8}$$

and therefore

$$\left(\frac{\vec{Y}^T\vec{Z} - \mu\vec{Y}^T\vec{1}}{2V_A\vec{Y}^T\mathbf{F}\vec{Y}}\right)^2 \sim \chi_1^2 \tag{9}$$

under the multivariate normal null. This fact can be used to test for an unexpectedly strong association between polygenic scores and a geographic axis by choosing $\vec{Y}$ to be the vector of latitudes or longitude across populations.

## tSDS vs. GWAS significance

### Polarizing SDS into tSDS

To analyze tSDS as a function of GWAS p-value, we first divided SNPs into 5% minor allele frequency bins. We standardized SDS values—subtracted the mean and divided by the standard deviation—within each bin. While SDS values were already standardized in a similar manner by *Field et al. (2016)*, we re-standardized SDS because the post-filtering composition of SNPs after in each GWAS was variable across GWAS. We then assigned tSDS values to each SNP by polarizing SDS to the tall allele. In other words, we set

$$tSDS := \begin{cases} SDS, & \text{derived} = \text{tall} \\ -SDS, & \text{otherwise} \end{cases} \tag{10}$$

where *derived* is the derived allele in UK10K (by which SDS was polarized in *Field et al., 2016*), and *tall* is the height increasing allele in the GWAS. We only used sites for which SDS values are available. Notably, this implicitly means that sites with minor allele frequency lower than 5% in UK10K were filtered out, due to the filtering used in *Field et al. (2016)*.

### Assessing significance of the correlation between GWAS p-value and tSDS

*Figure 2* illustrates the correlation between tSDS and GWAS p-value (p-value for the strength of association with height). We assessed the significance of the correlation between the two while accounting for LD between SNPs. To do this, we used a blocked-jackknife approach (*Kunsch, 1989*; *Busing et al., 1999*) to estimate the standard error of our Spearman correlation point estimate, $\hat{\rho}$. For each GWAS, SNPs were assigned to one of b = 200 contiguous blocks based on concatenated genomic coordinates. tSDS values should not be correlated across such large blocks. For each block $i$, we computed the Spearman correlation in the $i$'th jackknife sample, $\hat{\rho}_{(-i)}^b$—that is the Spearman correlation across all SNPs but the SNPs in block $i$. We then estimated the standard error of the point Spearman estimate by $\hat{\sigma}$, where

$$\hat{\sigma}^2 = \frac{b-1}{b}\sum_{i=1}^{b}(\hat{\rho}_{(-i)}^b - \bar{\rho}^b),$$

and.

$$\bar{\rho}^b = \frac{1}{b}\sum_{i=1}^{b}\hat{\rho}_{(-i)}^b$$

is the average of jackknife samples. Finally, we compute a p-value for the null hypothesis.

$$H_0 : \rho = 0,$$

by approximating $\hat{\rho}$ as Normally distributed under the null with standard deviation $\hat{\sigma}$, namely.

$$\hat{\rho} \sim N(0, \hat{\sigma}).$$

## LD Score regression

Summary statistics for traits were filtered and allele flipped using `munge_sumstats.py` (a python program provided by *Bulik-Sullivan et al., 2015b*), with the default filters. All regressions were performed using the LD Score Regression package, using the LD Scores derived from the 489 unrelated European individuals in 1000 Genomes Phase III and a modified SNP set that excluded the HLA, LCT, and chromosome eight inversion loci.

For genetic correlations of traits presented in *Table 2*, raw summary statistics were used. For other analyses, effect sizes of SNPs within each 1% minor allele frequency bin (as estimated by the 489 Europeans) were normalized to mean 0 and standard deviation 1, and those normalized statistics were used for downstream analyses. The standard two-step regression method from LD Score regression was used, with the default of 200 jackknife bins and a chi-square cutoff of 30, though results with UKB-GB were reasonably robust to a wide range of bin sizes and cutoffs.

## Data availability statement

The GWAS generated from the UK Biobank for this paper have been uploaded to Dryad: https://doi.org/10.5061/dryad.mg1rr36

The study also makes use of various publicly available GWAS datasets:

The data from the GIANT consortium GWAS, conducted by *Wood et al. (2014)* is available at the GIANT consortium website: https://portals.broadinstitute.org/collaboration/giant/index.php/GIANT_consortium_data_files

The UK Biobank GWAS of individuals of 'White British' ancestry only (UKB-GB), conducted by *Churchhouse and Neale (2017)*, is available at:http://www.nealelab.is/blog/2017/7/19/rapid-gwas-of-thousands-of-phenotypes-for-337000-samples-in-the-uk-biobank

The UK Biobank GWAS of individuals of broadly European ancestral (UKB-Eur), conducted by *Loh et al. (2017)*, is available at: https://data.broadinstitute.org/alkesgroup/UKBB/

Sibling GWAS data from *Robinson et al. (2015)* released in 2016 and now known to have been impacted by a computational bug, (R15-sibs, (*Robinson et al., 2016*)) as well as the newly rerun 2018 data (R15-sibs-updated, (*Robinson and Visscher, 2018*)) are both available at: http://cnsgenomics.com/data.html

Copies of these datasets are independently archived at: https://github.com/jjberg2/height-data.

## Acknowledgements

This research has been conducted using the UK Biobank resource under applications Number 11138 and 24983. We thank Manuel Rivas for assistance with the accessing this resource. We thank the Coop lab, Pritchard lab, Przeworski lab, Sella lab, Doc Edge, John Novembre, Guy Sella, Molly Przeworski, Joshua Schraiber, Loic Yengo and the authors of *Robinson et al. (2015)* and *Sohail et al., 2019* for helpful conversations and feedback on earlier drafts. We also thank Magnus Nordborg, Nicholas Barton, and Joachim Hermisson for helpful comments during the review process. JJB was supported in part by NIH R01 GM115889 to Guy Sella and in part by NIH Grant F32 GM126787 to JJB. AH was supported, in part, by a fellowship from the Stanford Center for Computational, Evolutionary and Human Genomics. NS-A was supported by a Stanford Graduate Fellowship and by the Department of Defense through a National Defense Science and Engineering Grant. HM was supported by NIH R01 GM121372 to Molly Przeworski. FR was supported by a Villum Fonden Young Investigator award. GC was partially supported by NIH R01 GM108779. Work in the Pritchard lab was supported in part by NIH R01 HG008140.

## Additional information

### Funding

| Funder | Grant reference number | Author |
|---|---|---|
| National Institutes of Health | R01 GM115889 | Jeremy J Berg |
| National Institutes of Health | F32 GM126787 | Jeremy J Berg |
| Stanford Center for Computational, Evolutionary and Human Genomics | Fellowship | Arbel Harpak |
| U.S. Department of Defense | National Defense Science and Engineering Gran | Nasa Sinnott-Armstrong |
| Stanford University | Graduate Fellowship | Nasa Sinnott-Armstrong |

| U.S. Department of Defense | National Defense Science and Engineering Grant | Nasa Sinnott-Armstrong |
| --- | --- | --- |
| National Institutes of Health | R01 GM121372 | Hakhamanesh Mostafavi |
| Villum Fonden | Young Investigator award | Fernando Racimo |
| National Institutes of Health | R01 HG008140 | Jonathan K Pritchard |
| National Institutes of Health | R01 GM108779 | Graham Coop |

The funders had no role in study design, data collection and interpretation, or the decision to submit the work for publication.

### Author contributions
Jeremy J Berg, Arbel Harpak, Nasa Sinnott-Armstrong, Conceptualization, Data curation, Software, Formal analysis, Investigation, Visualization, Methodology, Writing—original draft, Writing—review and editing; Anja Moltke Joergensen, Validation, Investigation, Visualization, Writing—original draft; Hakhamanesh Mostafavi, Data curation, Methodology, Writing—review and editing; Yair Field, Evan August Boyle, Xinjun Zhang, Conceptualization; Fernando Racimo, Conceptualization, Supervision, Investigation, Writing—original draft, Project administration, Writing—review and editing; Jonathan K Pritchard, Conceptualization, Supervision, Writing—original draft, Project administration, Writing—review and editing; Graham Coop, Conceptualization, Supervision, Project administration

### Author ORCIDs
Jeremy J Berg http://orcid.org/0000-0001-5411-6840
Arbel Harpak https://orcid.org/0000-0002-3655-748X
Nasa Sinnott-Armstrong http://orcid.org/0000-0003-4490-0601
Hakhamanesh Mostafavi http://orcid.org/0000-0002-1060-2844
Evan August Boyle https://orcid.org/0000-0003-4494-9771
Xinjun Zhang https://orcid.org/0000-0003-1298-3545
Fernando Racimo https://orcid.org/0000-0002-5025-2607
Jonathan K Pritchard https://orcid.org/0000-0002-8828-5236
Graham Coop http://orcid.org/0000-0001-8431-0302

### Decision letter and Author response
Decision letter https://doi.org/10.7554/eLife.39725.050
Author response https://doi.org/10.7554/eLife.39725.051

## Additional files

### Supplementary files
• Transparent reporting form
DOI: https://doi.org/10.7554/eLife.39725.017

### Data availability
The GWAS generated from the UK Biobank for this paper have been uploaded to Dryad:. The study also makes use of various publicly available GWAS datasets. The data from the GIANT consortium GWAS, conducted by Wood et al. (2014) is available at the GIANT consortium website: https://portals.broadinstitute.org/collaboration/giant/index.php/GIANT_ consortium_data_1les. The UK Biobank GWAS of individuals of "White British" ancestry only (UKB-GB), conducted by Churchhouse et al. (2017), is available at: http://www.nealelab.is/blog/2017/7/19/rapid-gwas-of-thousands-ofphe-notypes- for-337000-samples-in-the-uk-biobank. The UK Biobank GWAS of individuals of broadly European ancestral (UKB-Eur), conducted by Loh et al. (2017), is available at: https://data.broadinstitute.org/alkesgroup/UKBB/. Sibling GWAS data from Robinson et al. (2015) released in 2016 and now known to have been impacted by a computational bug, (R15-sibs, Robinson et al. (2016)) as well as the newly rerun 2018 data (R15-sibs-updated, Robinson and Visscher (2018)) are both available at: http://cnsgenomics.com/data.html. Copies of these datasets are archived at: https://github.com/jjberg2/height-data.

The following dataset was generated:

| Author(s) | Year | Dataset title | Dataset URL | Database and Identifier |
|---|---|---|---|---|
| Jeremy J Berg, Arbel Harpak, Nasa Sinnott-Armstrong, Anja Moltke Joergensen, Hakhamanesh Mostafavi, Yair Field, Evan August Boyle | 2019 | Data from: Reduced signal for polygenic adaptation of height in UK Biobank | http://dx.doi.org/10.5061/dryad.mg1rr36 | Dryad Digital Repository, 10.5061/dryad.mg1rr36 |

The following previously published datasets were used:

| Author(s) | Year | Dataset title | Dataset URL | Database and Identifier |
|---|---|---|---|---|
| Wood AR, Esko T, Yang J, Vedantam S, Pers TH, Gustafsson S | 2014 | GIANT Consortium 2014 Height Summary Statistics | https://portals.broadinstitute.org/collaboration/giant/index.php/GIANT_consortium_data_files#GWAS_Anthropometric_2014_Height | GIANT Consortium Website, GWAS_Anthropometric_2014_Height |
| Loh PR, Kichaev G, Gazal S, Schoech AP, Price A | 2017 | Price Lab UK Biobank GWAS | https://data.broadinstitute.org/alkesgroup/UKBB/ | Broad Institute Website, UKBB |
| Robinson R | 2016 | Robinson and Visscher 2016 Height Summary Statistics | http://cnsgenomics.com/data/robinson_et_al_2015_ng/withinfam_summary_ht_bmi_release_March2016.tar.gz | University of Queensland Program in Complex Trait Genetics Website, withinfam_summary_ht_bmi_release_March2016.tar.gz |
| Robinson R | 2018 | Robinson and Visscher 2018 Corrected Height Summary Statistics | http://cnsgenomics.com/data/robinson_et_al_2015_ng/Within-family_GWAS_of_height_based_on_sib_regression_using_data_from_Robinson_et_al_2015_LYMRR.txt.gz | University of Queensland Program in Complex Trait Genetics Website, Within-family_GWAS_of_height_based_on_sib_regression_using_data_from_Robinson_et_al_2015_LYMRR.txt.gz |

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

## Appendix 1

DOI: https://doi.org/10.7554/eLife.39725.018

## Expanded SNP Sets

Some analyses of polygenic score variation among populations have used many more than the SNPs we use in our main text analyses, in the hope of increasing power to detect adaptive divergence (e.g. *Robinson et al., 2015*). Here, we use three alternative ascertainment schemes that increase the number of SNPs used, and apply them to the UKB-GB GWAS to determine the resulting effect on the signal of selection:

### 20k

19,848 genotyped SNPs ascertained from the UKB-GB dataset by running plink's clumping procedure with $r^2 < 0.1$, a maximum clump size of 1Mb, $p < 0.01$, and using 10,000 randomly selected unrelated British ancestry individuals as the reference for LD structure.

### 5k

4,880 SNPs with the smallest p values subsampled from the 20k ascertainment.

### HapMap5k

5,675 SNPs ascertained from UKB-GB GWAS SNPs after first restricting to HapMap3 SNPs (*International HapMap 3 Consortium, 2010*), using the same plink clumping procedure as the 20k ascertainment. This HapMap3 ascertainment was performed in order to mimic the ascertainment in *Robinson et al. (2015)*.

We also tested two alternative ascertainments of the R15-sibs-updated dataset, described immediately below. While the majority of this appendix focuses on analyses of the three alternate ascertainments of the UKB-GB datasets, described above, we include a brief analysis of these R15-sibs-updated alternate ascertainments.

### R15-sibs-updated-3.5k

3,579 SNPs ascertained from the R15-sibs-updated dataset by running plink's clumping procedure with $r^2 < 0.1$ a maximum clump size of 1Mb, $p < 0.01$, using the same set of 10,000 randomly selected unrelated British ancestry individuals as the reference for LD structure.

### R15-sibs-updated-22k

22,243 SNPs ascertained under the same plink setting as R15-sibs-updated-3.5k, but with the p value threshold relaxed to $p < 0.1$.

For each expanded SNP set, we applied both the general $Q_X$ test for overdispersion, as well as the specific test for a correlation with latitude (both tests are outlined in the Materials and methods). In all three datasets, the relationship between polygenic scores and latitude was consistent with neutrality. However, in both the 20k and HapMap5k datasets, we can reject the neutral model, as the $Q_X$ p value is $1.68 \times 10^{-3}$ for 20k and $9.88 \times 10^{-9}$ for HapMap5k. On the other hand, 5k is not significant, with a p value of 0.75.

We were concerned that the rejection of the neutral null with 20k and HapMap5k ascertainments may be partly due to the higher proximity of SNPs included—leading to deviations from the independent evolution assumption of the neutral model underlying the $Q_X$ hypothesis test. To investigate this, we leveraged a decomposition of the $Q_X$ statistic in terms of the underlying loci used to calculate the polygenic scores. Specifically, *Berg and Coop (2014)* showed that $Q_X$ can be expressed in terms of and '$F_{ST}$-like' component, which

describes the extent to which loci underlying the polygenic scores are marginally overdispersed, and an 'LD-like' component, which describes the extent to which pairs of loci which affect the trait covary in their allele frequencies across populations. This decomposition can be written as

$$Q_X = \underbrace{(M-1)\frac{2\sum_\ell \alpha_\ell^2 Var(\vec{p}_\ell)}{V_A}}_{F_{ST}-\text{like term}} + \underbrace{(M-1)\frac{2\sum_{\ell \neq \ell'} \alpha_\ell \alpha_{\ell'} Cov(\vec{p}_\ell, \vec{p}_{\ell'})}{V_A}}_{LD-\text{like term}}. \tag{A1}$$

Here, we have assumed that the allele frequencies, $p_l$, have been transformed so as to remove the influence of population structure. See the discussion surrounding equations 12-14 in **Berg and Coop (2014)** for a more complete explanation of this transformation.

Here, we extend this decomposition further, breaking the LD-like term into components as a function of the degree of physical separation of SNPs along the chromsome. Specifically, we define a set of partial $Q_X$ statistics ($pQ_X(k)$), such that $pQ_X(k)$ gives the contribution to from sites which are $k$ SNPs apart on the chromosome:

$$pQ_X(k) = (M-1)\frac{2\sum_{\ell,\ell' \in \mathbb{S}_k} \alpha_\ell \alpha_{\ell'} Cov(\vec{p}_\ell, \vec{p}_{\ell'})}{V_A} \tag{A2}$$

where $\mathbb{S}_k$ denotes the set of SNP pairs which are $k$ SNPs apart on the same chromosome (note that only SNPs included in the a given ascertainment are included for the purposes of counting how many SNPs apart any two SNPs are). So $pQ_X(0)$ would give the ' $F_{ST}$ term', while $pQ_X(1)$ gives the component of the 'LD term' that comes from covariance between pairs of SNPs which do not have another SNP (that is included in the polygenic scores) physically located between them. $pQ_X(2)$ gives the component that comes from covariance between pairs of SNPs separated by exactly one other SNP included in the polygenic scores, $pQ_X(3)$ the component from SNPs separated by exactly two intervening SNPs, etc. We let $\mathbb{S}_\infty$ be the set of pairs which are on separate chromosomes, so that $pQ_X(\infty)$ gives the contribution to $Q_X$ coming from pairs of SNPs on different chromosomes. This decomposition retains the property that

$$Q_X = \sum_{k=0}^{K_{max}} pQ_X(k) + pQ_X(\infty), \tag{A3}$$

where $K_{max}$ is the maximum separation of two SNPs on any chromosome. We note that the $pQ_X(k)$ terms are not independent of one another, but they are uncorrelated under the neutral null.

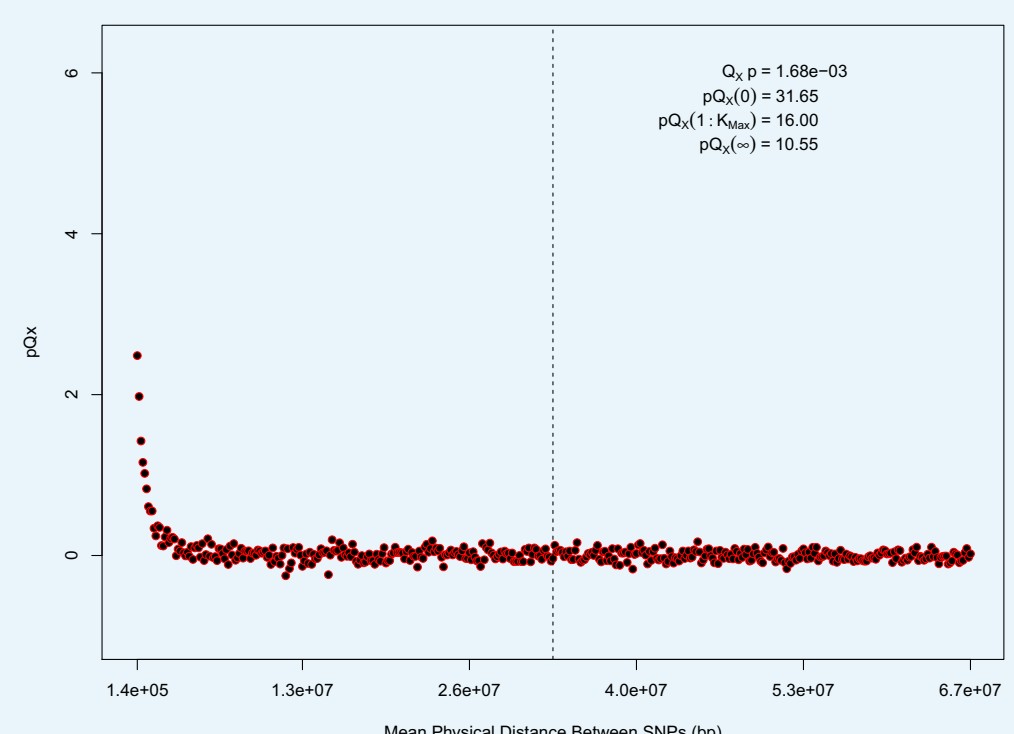

**Appendix 1—figure 1.** $pQ_X(k)$ statistics for $k = 1:450$ for the 20 k dataset. The x axis gives the average physical distance between all pairs of SNPs contributing to a given $pQ_X(k)$ statistic. The uptick in $pQ_X(k)$ on the left side of the plot (i.e. small values of $k$) indicates that SNPs which are physically close to one another and have the same sign in their effect on height covary across population disproportionately as compared to more distant pairs of SNPs. Note that the number of pairs of SNPs ($|\mathbb{S}_k|$) contributing to a given $pQ_X(k)$ decreases as $k$ increases, as smaller chromosomes have fewer pairs at larger distances than they do at shorter distances. This leads to a decrease in the variance of $pQ_X(k)$ under the null as $k$ increases. However, this decline in variance is not responsible for the decay in signal as $k$ increases, as $|\mathbb{S}_k|$ remains approximately constant until well past the dashed vertical line, which indicates the distance between between the ends of chromosome 21 (the shortest chromosome, and therefore the first to drop out of the $pQ_X(k)$ calculation).

DOI: https://doi.org/10.7554/eLife.39725.019

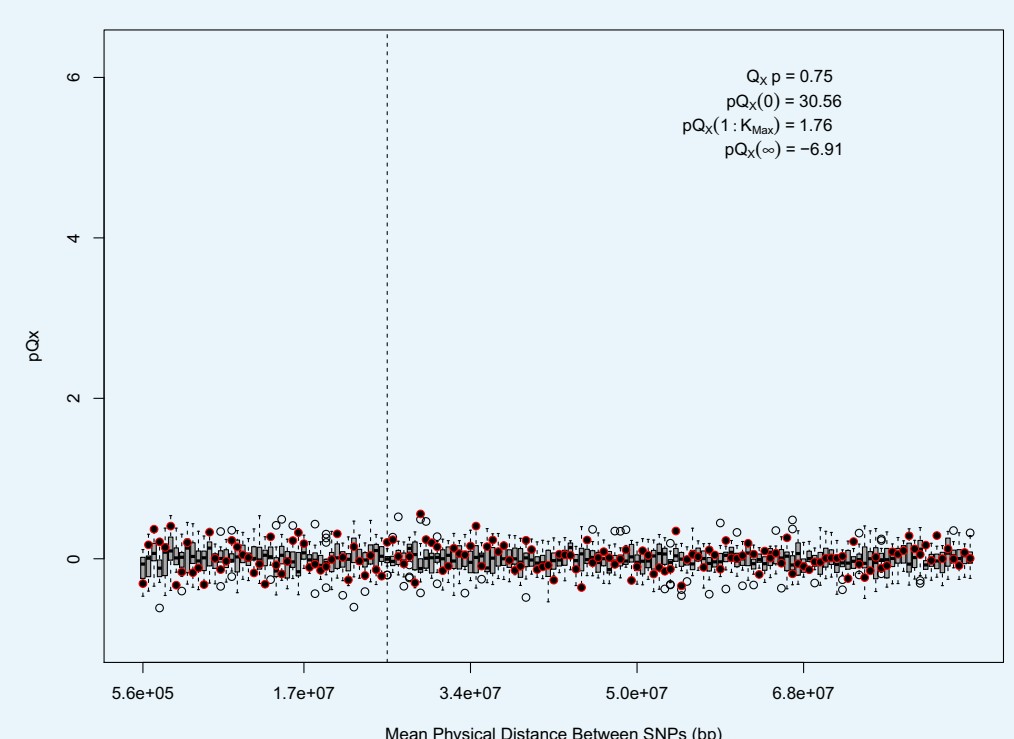

**Appendix 1—figure 2.** $pQ_X(k)$ statistics for $k = 1 : 150$ for the 5 k dataset. The x axis gives the average physical distance between all pairs of SNPs contributing to a given $pQ_X(k)$ statistic. The boxplots give an empirical null distribution of $pQ_X(k)$ statistics derived from permuting the signs of all effect sizes independently (this empirical null was omitted from **Appendix 1—figure 1** due to computational expense). In this case, SNPs that are physically close to one another do not contribute disproportionately to the signal.

DOI: https://doi.org/10.7554/eLife.39725.020

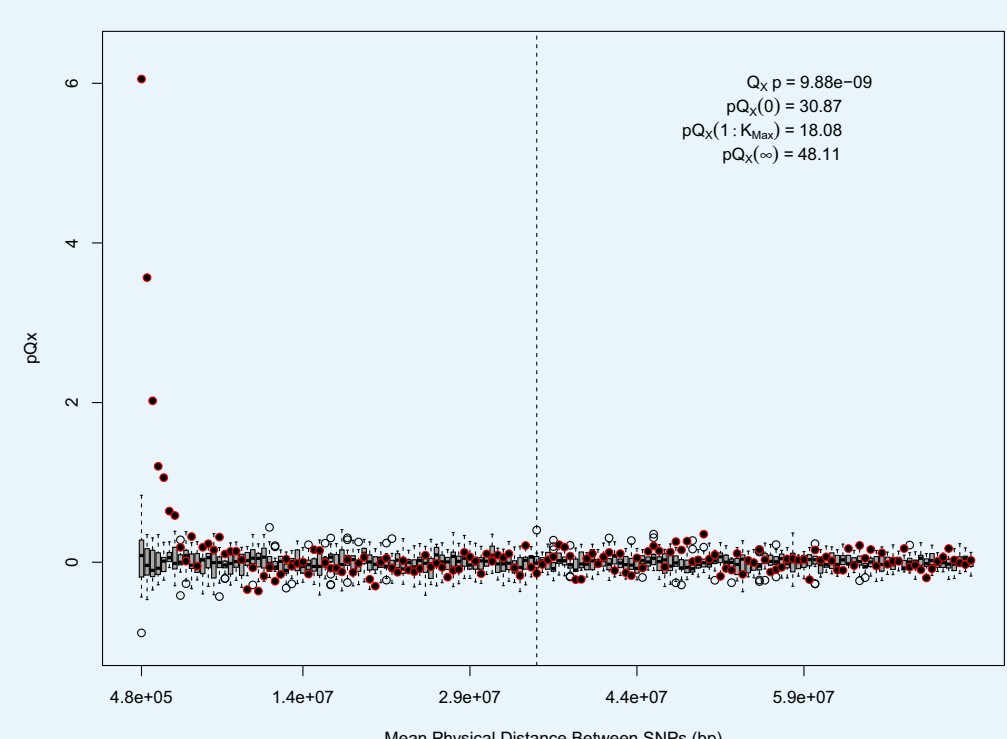

**Appendix 1—figure 3.** $pQ_X(k)$ statistics for $k = 1 : 150$ for the HapMap5k dataset. The x axis gives the average physical distance between all pairs of SNPs contributing to a given $pQ_X(k)$ statistic. The boxplots give an empirical null distribution of $pQ_X(k)$ statistics derived from permuting the signs of all effect sizes independently (this empirical null was omitted from *Appendix 1—figure 1* due to computational expense). The uptick in signal from pairs of SNPs physically nearby to one another is present in this dataset, again suggesting a role for physical linkage in contributing to the signal. However, note that in contrast to the 20 k and 5 k ascertainments, the HapMap5k ascertainment also has a large amount of signal from $pQ_X(\infty)$, which cannot be explained by linkage.

DOI: https://doi.org/10.7554/eLife.39725.021

In *Appendix 1—figure 4* and *5* of this Appendix, we show the pQx statistics for various $k$ values in these three different ascertainments. In both the 20 k and the HapMap5k ascertainments, $pQ_X$ is higher for low $k$ values–that is there is more signal coming from covariance among SNP pairs which are physically close to one another on the chromosome than from distant pairs. This indicates a role for linkage in generating the signals detected in these two ascertainments (we also observed these sort of signals in the R15-sibs-updated-3.5k and R15-sibs-updated-22k ascertainments; *Appendix 1—figure 4* and *5*). In contrast, we see no linkage-associated signal in the 5 k ascertainment (in fact, we see no signal whatsoever). The major difference between the signal we observe in the 20 k ascertainment and that in the HapMap5k ascertainment is that $pQ_X(\infty)$ is strongly positive for the HapMap5k ascertainment, whereas it is weakly negative for the 20 k ascertainment. This difference in the strength of between-population LD between loci on separate chromosomes is largely responsible for the fact that the neutral null hypothesis is strongly rejected for the 20 k ascertainment, but only weakly so for the HapMap5k ascertainment.

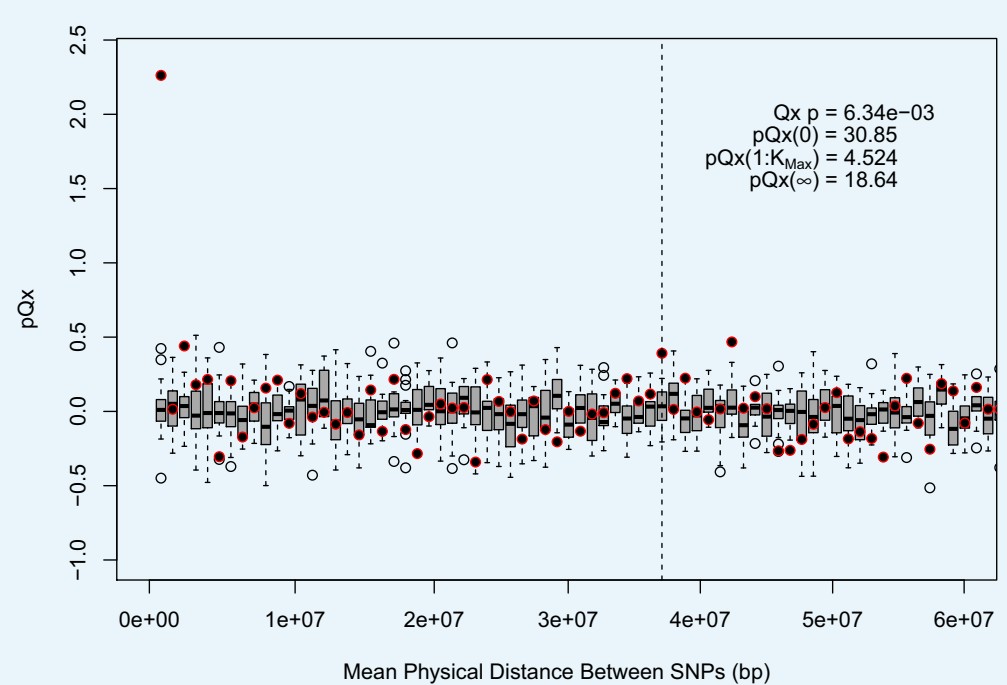

**Appendix 1—figure 4.** $pQ_X(k)$ statistics for the R15-sibs-updated-3.5k ascertainment. Similar to the expanded UKB-GB ascertainments, the elevated signal from covariance among SNPs in adjacent bins suggests that the independence assumption of the neutral model is being violated.

DOI: https://doi.org/10.7554/eLife.39725.022

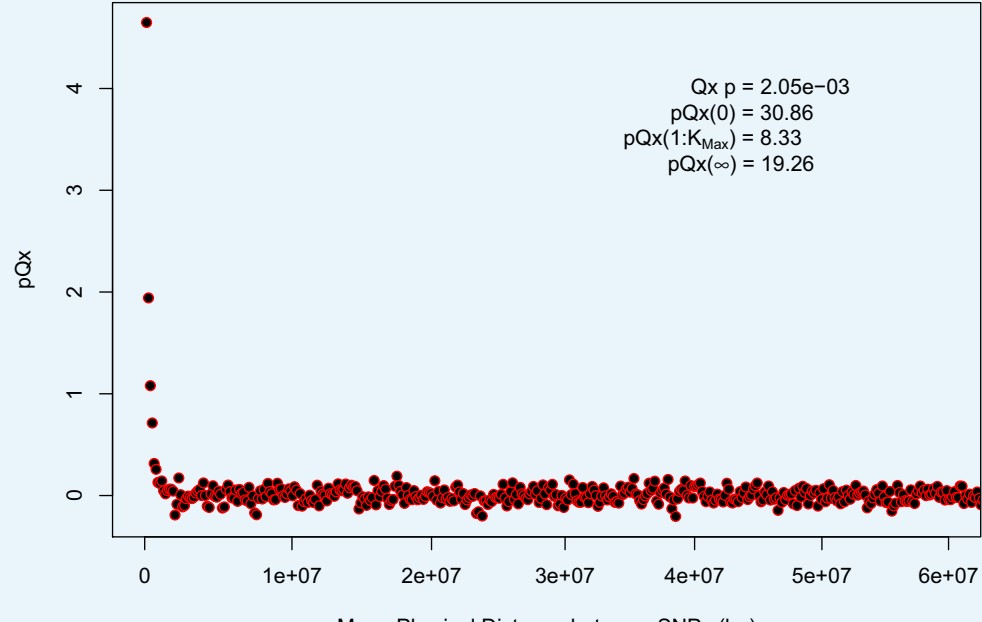

**Appendix 1—figure 5.** $pQ_X(k)$ statistics for the R15-sibs-updated-22k. Similar to **Appendix 1—figure 4**, the elevated signal from covariance among SNPs in adjacent or nearby bins suggests that the independence assumption of the neutral model is being violated. Similar to **Appendix 1—figure 1**, we omitted the sign flipping null due to computational expense.

DOI: https://doi.org/10.7554/eLife.39725.023

This heterogeneity of signals across different ascertainments suggests that the signals we do observe are unlikely to be the result of selection—but rather result from some other process or phenomenon which we do not fully understand. Perhaps the most unusual observation is the fact that the among chromosome component of $Q_X$ (i.e. $pQ_X(\infty)$) is so strong from HapMap5k, when it is absent under all other ascertainments. This suggests a role for some ascertainment bias impacting SNPs included in the HapMap3 SNP set. This seems plausible, as SNPs included in the HapMap3 SNP set have an elevated minor allele frequency as compared to a genome-wide sample. While it seems plausible that patterns of among population LD would differ for SNPs included on genotyping platforms, it is not clear why among-population LD should be systematically positive with respect to the SNPs' effect on height.

To better understand the signal observed in the HapMap5k ascertainment, we make use of an alternate decomposition of the $Q_X$ statistic (**Berg et al., 2017**; **Josephs et al., 2018**). First, write the eigenvector decomposition of $\mathbf{F}$ as $\mathbf{U}\Lambda\mathbf{U^T}$. The $m^{th}$ column of $\mathbf{U}$ ($\vec{U}_m$) gives the $m^{th}$ eigenvector of $\mathbf{F}$, and the $m^{th}$ diagonal entry of $\Lambda$ ($\lambda_m$) gives the $m^{th}$ eigenvalue of $\mathbf{F}$. Note that because this eigen-decomposition is performed on the population level covariance matrix, they capture only the major axes of variation among our pre-specified population labels, in contrast to how PCA is usually done at the individual level in demographic inference applications. Now, we can define a statistic

$$Q_U(m) = \frac{\left(\left(\vec{Z}-\mu\right)^T \vec{U}_m\right)^2}{2\lambda_m V_A} \tag{A4}$$

which has a $\chi_1^2$ distribution under the neutral null hypothesis. These statistics, like the $pQ_X$ statistics, have the property that $Q_X$ is given simply by their sum:

$$Q_X = \frac{\left(\vec{Z}-\mu\right)^T \mathbf{F}^{-1}\left(\vec{Z}-\mu\right)}{2V_A} \tag{A5}$$

$$= \frac{\left(\vec{Z}-\mu\right)^T \mathbf{U}\Lambda^{-1}\mathbf{U}^T\left(\vec{Z}-\mu\right)}{2V_A} \tag{A6}$$

$$= \sum_m \frac{\left(\left(\vec{Z}-\mu\right)^T \vec{U}_m\right)^2}{2\lambda_m V_A} \tag{A7}$$

$$= \sum_m Q_U(m). \tag{A8}$$

An unusually large value of $Q_U(m)$ for a given choice of $m$ is an indication that the polygenic scores are more strongly correlated with the $m^{th}$ axis of population structure than expected under the neutral null model. Therefore, once a signal is detected with $Q_X$, the $Q_U$ statistics can be used to understand which specific axes of divergence among populations are responsible for generating the signal in $Q_X$.

In **Appendix—figure 6**, we show a quantile-quantile plot of the $-log_{10}$ p values for the HapMap5k ascertainment, derived from comparing these $Q_U$ statistics from the European set of populations to the $\chi_1^2$ distribution. It is particularly noteworthy that the signal in this ascertainment is diffuse, resulting from inflation of nearly all of the $Q_U$ statistics, rather than just a few. This is a statement that the signal detected in the HapMap5k ascertainment results from the polygenic scores simply being more variable along all axes, rather than one particular axis of population structure. In general, we are skeptical that this represents a real signal of selection, particularly given how sensitive it is to ascertainment.

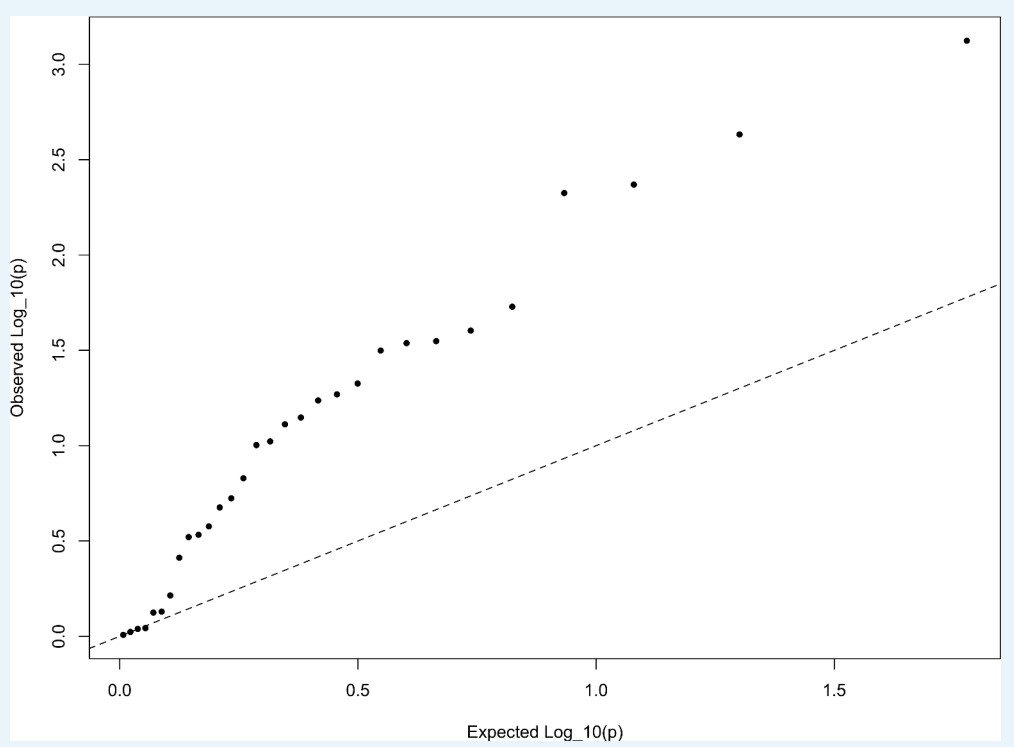

**Appendix 1—figure 6.** The QQ plot of $-log_{10}$ P values for the $Q_U$ statistics calculated a within-Europe sample using the HapMap5k ascertainment. The systematic inflation of $Q_U$ indicates a non-specific rejection of neutrality: polygenic scores are more variable in all directions than expected under the null model. This pattern is not expected under adaptive divergence of labeled populations.

DOI: https://doi.org/10.7554/eLife.39725.024

One biological hypothesis is that the HapMap5k ascertainment could suggest ancient assortative mating on the basis of height. Specifically, our neutral null model assumes that all loci drift independently. However, assortative mating on the basis of a phenotype will lead to a build-up of within population LD that is positive with respect to the direction of allelic effects on the trait—even among distant or unlinked alleles. As populations drifted apart, within-population LD due to assortative mating would get converted into among-population LD—causing a deviation from our null-model assumption of independent evolution across all loci. This phenomenon would result in populations drifting apart in height-associated loci faster than expected by the rest of the genome. This hypothesis is consistent with the diffusion of $Q_X$ across all $Q_U$ terms in HapMap5k. This hypothesis is also consistent with higher $pQ_X$ for physically proximate SNPs, as assortative mating would leads to a stronger buildup of trait LD among pairs of loci which are tightly linked than for those that are not, which would lead to stronger among population LD among these loci as populations diverge

However, under this hypothesis, it is not clear why we would expect the uptick in $pQx(k)$ for small k to be present in the HapMap5k and 20 k datasets but not the 5 k dataset, or why the $pQx(\infty)$ signal should be present in only the HapMap5k dataset. At this point, we leave the assortative mating hypothesis outlined above as purely speculative, and leave further investigations for future work.

It also possible that all of the signals seen here are entirely the result of a violation of the assumption of independence among SNPs under the null model. This may be the case even for $pQx(\infty)$ signals. Consider for example 3 SNPs, with SNPs 1 and 2 adjacent to one another on a chromosome, and SNP three located on a different chromosome. $pQx(\infty)$ would include the covariance between SNP 1 and 3, and that between 2 and 3. However, if SNPs 1 and 2 covary, then these two covariance terms will be correlated with one another. Therefore, the variance of the $pQx(\infty)$ term will be underestimated by a null that assumes independence

among SNPs, even though all of the terms that contribute to it come from covariance among SNPs on separate chromosomes.

## Appendix 2

DOI: https://doi.org/10.7554/eLife.39725.018

# Robustness of differences in differentiation signal to filtering schemes

Here, we explore whether the failure to replicate GIANT signals with UKB-GB could be explained as a result of differences in filtering of SNPs in one or the other dataset. This analysis was performed by Anja Moltke Jørgensen and Fernando Racimo, and doubles as a demonstration of the failure to replicate the GIANT signal of excess among population variance (*Turchin et al., 2012*; *Berg and Coop, 2014*; *Robinson et al., 2015*; *Zoledziewska et al., 2015*; *Berg et al., 2017*) that was performed independently from that in the main text.

We focused on present-day populations from phase 3 of the 1000 Genomes Project (*1000 Genomes Project Consortium et al., 2015*). We divided the genome into 1700 approximately independent LD blocks, using fgwas (*Pickrell, 2014*; *Berisa and Pickrell, 2016*), and extracted, for each of the two GWAS for height, the SNP with the highest posterior probability of association (PPA) from each block, using. This resulted in a total of 1700 SNPs (one per block). Unless otherwise stated, we computed scores using the subset of these SNPs that were located in blocks with high per-block posterior probability of association ($PPA > 95\%$), after retrieving the allele frequencies of these SNPs in the 1000 Genomes population panels, using glactools (*Renaud, 2018*). We tested different types of filters to assess how they influenced the results.

*Appendix 2—figure 2* (upper row) shows that genetic scores computed for each of the 1000 Genomes phase three populations. In each plot below in which we report a P-value, this P-value comes from calculating the $Q_X$ statistic, and assuming this statistic is chi-squared distributed (*Berg and Coop, 2014*; *Berg et al., 2017*). The candidate SNPs used for calculating the genetic scores were filtered so that the average minor allele frequency across populations was more than or equal to $5\%$.

To investigate the effect of the per-block posterior probability of association (Block PPA) on the genetic scores, we also used two alternative PPA thresholds for including a block in the computation of the PPA score: 0 (i.e. including all blocks, lower row of *Appendix 2—figure 2*) and 0.5 (middle row of *Appendix 2—figure 2*) shows that this filtering has little effect in the difference in results between the two GWASs.

| Population.Code | Population.Description |
|---|---|
| ACB | African Caribbeans in Barbados |
| ASW | Americans of African Ancestry in SW USA |
| ESN | Esan in Nigeria |
| GWD | Gambian in Western Divisions in the Gambia |
| LWK | Luhya in Webuye, Kenya |
| MSL | Mende in Sierra Leone |
| YRI | Yoruba in Ibadan, Nigeria |
| CLM | Colombians from Medellin, Colombia |
| MXL | Mexican Ancestry from Los Angeles USA |
| PEL | Peruvians from Lima, Peru |
| PUR | Puerto Ricans from Puerto Rico |
| CDX | Chinese Dai in Xishuangbanna, China |
| CHB | Han Chinese in Beijing, China |
| CHS | Southern Han Chinese |
| JPT | Japanese in Tokyo, Japan |
| KHV | Kinh in Ho Chi Minh City, Vietnam |
| CEU | Utah Residents (CEPH) with Northern and Western European Ancestry |
| FIN | Finnish in Finland |
| GBR | British in England and Scotland |
| IBS | Iberian Population in Spain |
| TSI | Toscani in Italia |
| BEB | Bengali from Bangladesh |
| GIH | Gujarati Indian from Houston, Texas |
| ITU | Indian Telugu from the UK |
| PJL | Punjabi from Lahore, Pakistan |
| STU | Sri Lankan Tamil from the UK |

**Appendix 2—figure 1.** Present-day populations from 1000 Genomes Project Phase 3 used to build population-level polygenic scores, colored by their respective super-population code.

DOI: https://doi.org/10.7554/eLife.39725.026

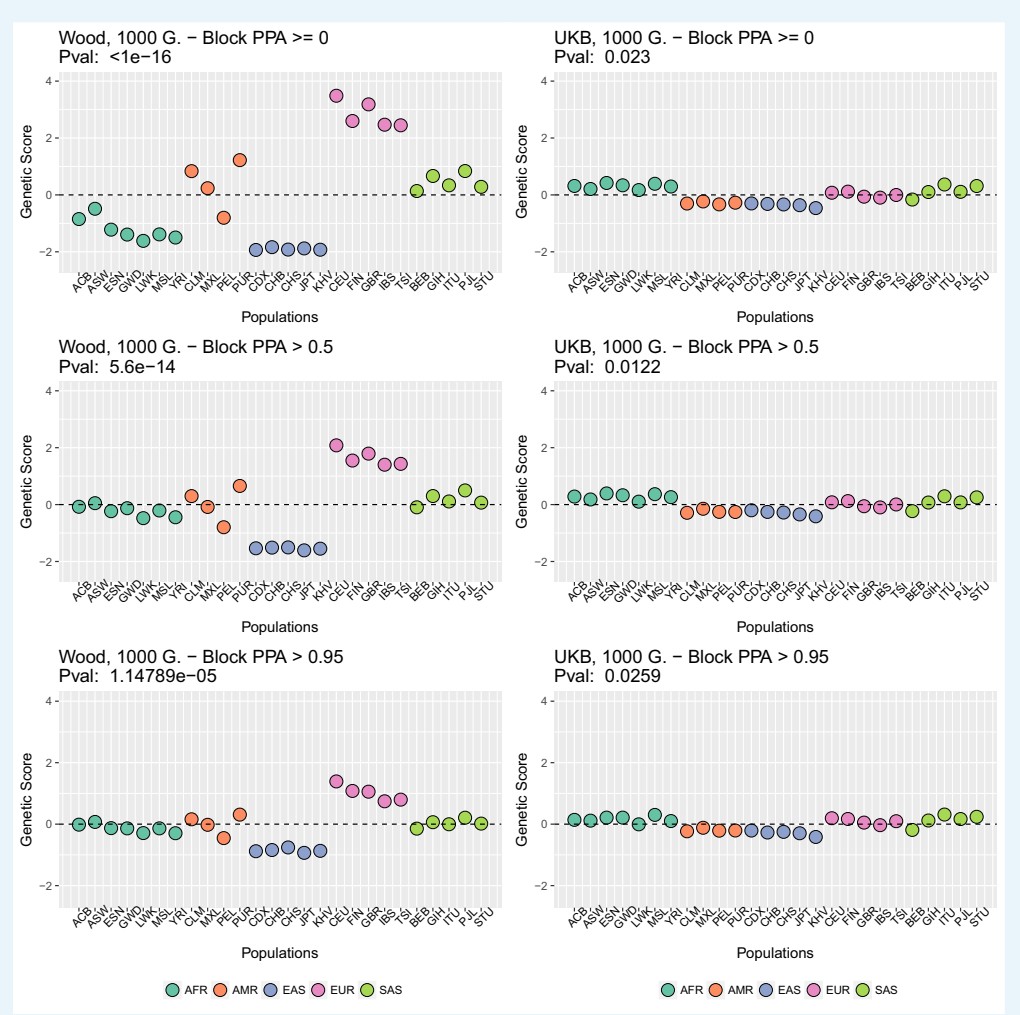

**Appendix 2—figure 2.** Genetic scores in present-day populations, colored by their super-population code, and created using different block-PPA thresholds. Left column: *Wood et al. (2014)* GWAS. Right column: Neale lab UK Biobank GWAS.

DOI: https://doi.org/10.7554/eLife.39725.027

To visualize the contribution of each SNP to the difference in scores between two populations with high differentiation in the Wood et al. GWAS (CHB and CEU), we produced a contour plot in which we display the absolute effect size of each SNP contributing in the computation of the genetics scores, plotted as a function of the difference in the frequency of the trait-increasing allele for that SNP in the two populations (*Appendix 2—figure 3*).

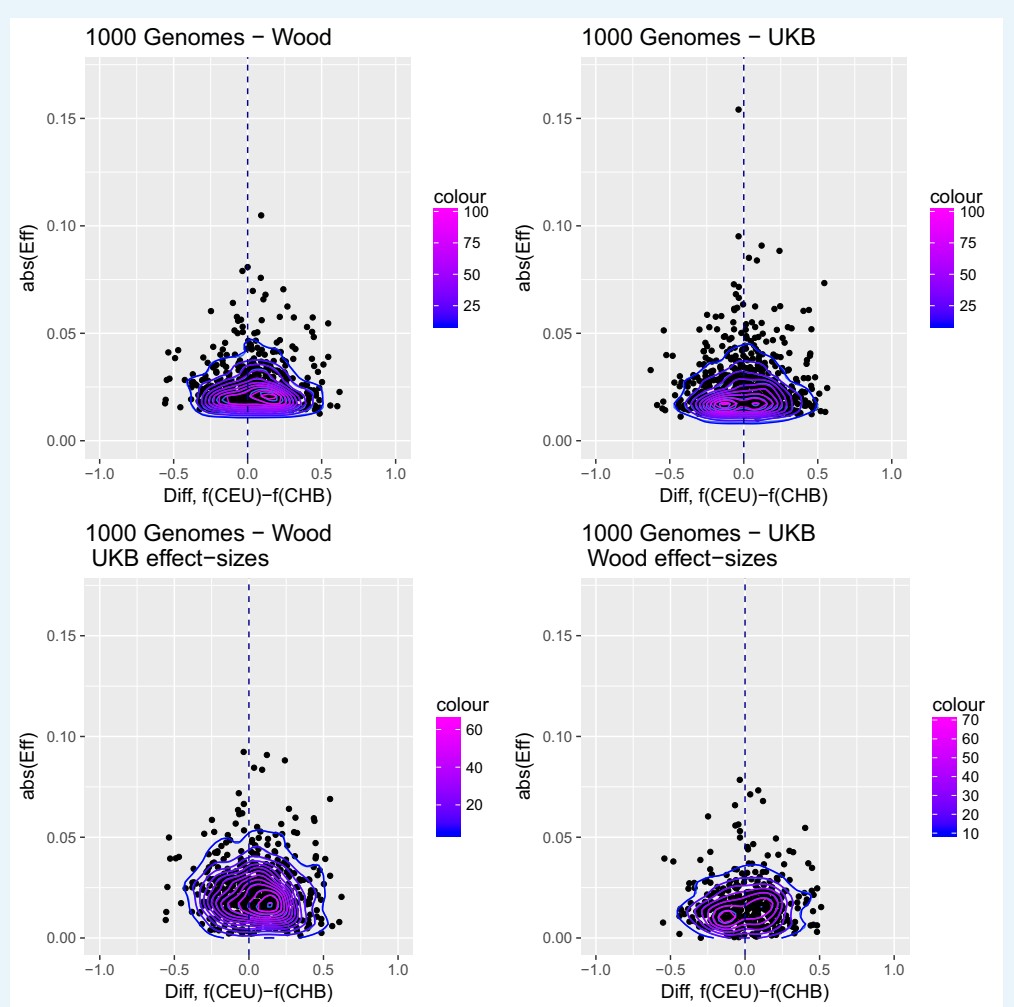

**Appendix 2—figure 3.** Distribution of the absolute value of effect sizes (y-axis) plotted as a function of the difference in frequency of the trait-increasing allele between CEU and CHB (x-axis), for candidate SNPs used to build genetic scores. Top left: trait-associated SNPs from Wood et al., with effect sizes from the same GWAS. Top right: trait-associated SNPs from the Neale lab GWAS, with effect sizes from the same GWAS. Bottom left: trait-associated SNPs from Wood et al., but with their corresponding effect sizes from the Neale lab GWAS. Bottom right: trait-associated SNPs from the Neale lab GWAS, but with their corresponding effect sizes from Wood et al. Contour colors denote the density of SNPs in different regions of each plot.

DOI: https://doi.org/10.7554/eLife.39725.028

*Appendix 2—figure 3* shows that the distribution of the difference in scores between the two populations is shifted in favor of CEU when using the Wood et al. dataset, but not when using the UKB dataset. When selecting SNPs via PPAs from the Wood et al. dataset but using their UKB effect sizes, the distribution of differences is also shifted in favor of CEU, but this does not occur when performing the converse: using PPAs from UKB to select SNPs, but plotting their effect sizes from Wood et al.

This figure also reveals that there are a number of SNPs in the UKB dataset with high effect sizes and very small differences in allele frequency between the two populations. These SNPs tend to have allele frequencies near the boundaries of extinction or fixation in both populations, suggesting they could possibly be under the influence of negative selection. To investigate the contribution of these high-effect SNPs on the overall genetic scores with the UKB dataset, we removed their corresponding blocks from the score computation, and re-

calculated the genetic scores for all populations. We chose a minimum absolute effect size equal to 0.08 for removal of SNPs, and the 6 SNPs in the UKB dataset which are above this threshold were therefore excluded from the analysis. This filtering, however, does not seem to serve to recover the Wood et al. signal (*Appendix 2—figure 4*).

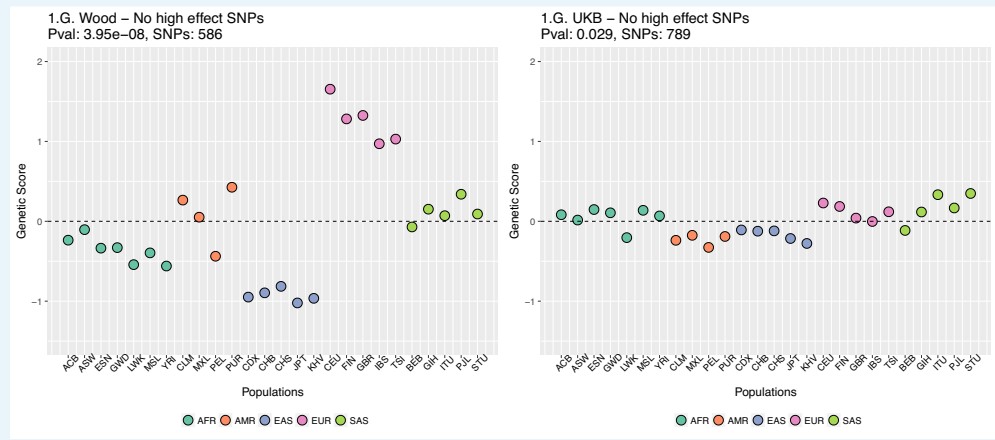

**Appendix 2—figure 4.** Genetic scores for present-day populations, after excluding 6 high-effect SNPs from UKB, colored by super-population code. Left: Wood et al. GWAS. Right: Neale lab UK Biobank GWAS.

DOI: https://doi.org/10.7554/eLife.39725.029

In *Appendix 2—figure 5* we restrict the candidate SNPs used, by only allowing SNPs that have minor allele frequencies larger than 0.05 in all populations. This is different from our previous default allele frequency filtering, in which we only required the average of the minor allele frequency across populations to be larger than 0.05. Nevertheless, this filtering does not recover the Wood et al. signal either.

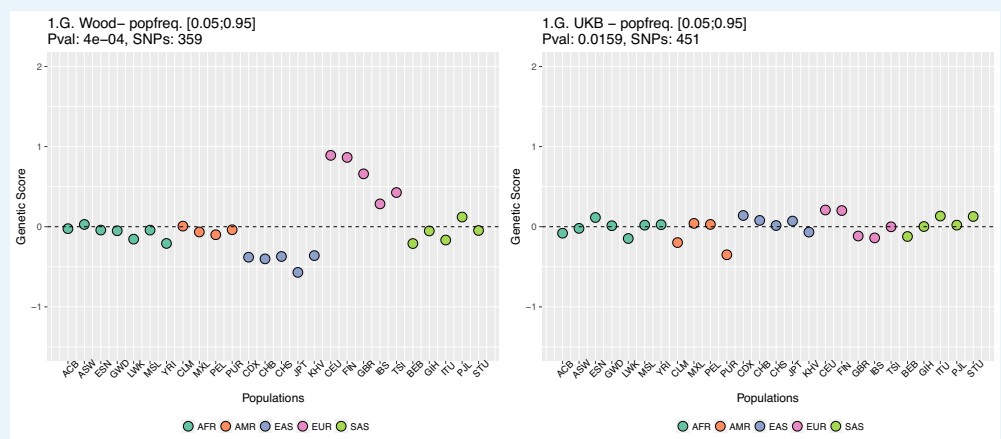

**Appendix 2—figure 5.** Genetic scores computed only with SNPs that have minor allele frequencies larger than 0.05 in all populations. Left: Wood et al. GWAS. Right: Neale lab UK Biobank GWAS.

DOI: https://doi.org/10.7554/eLife.39725.030

We also looked into whether the candidate SNPs found using the UK Biobank dataset were also present in the Wood et al. GWAS, but perhaps with much smaller effect sizes, and this was somehow affecting the genetic scores made using the UKB data. In *Appendix 2—figure 6* all UK Biobank candidate SNPs that were also found in Wood et al. were evaluated and if a

SNP's absolute effect size in Wood et al. was smaller than or equal to 0.05, the SNP was excluded from the UK Biobank candidate set.

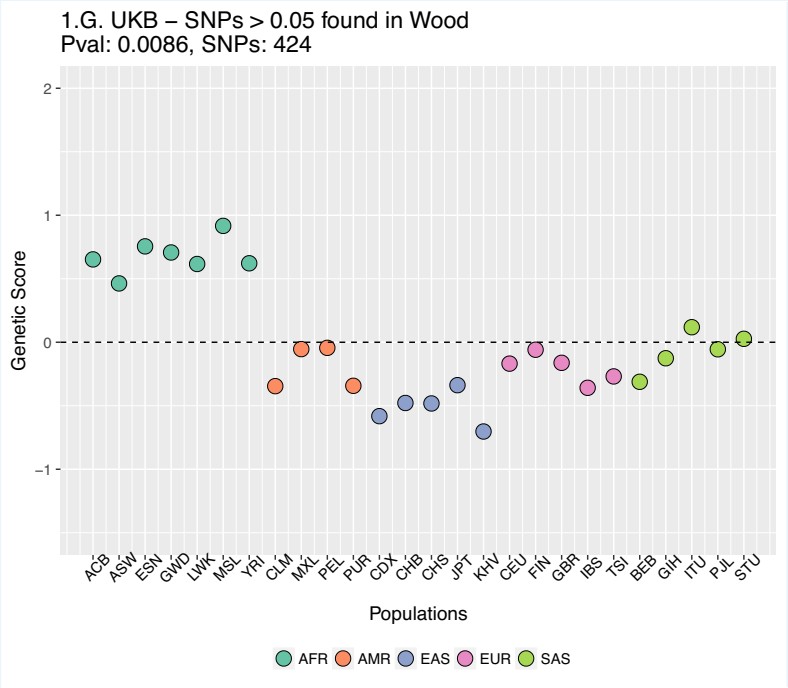

**Appendix 2—figure 6.** Genetic scores computed using the UK Biobank data, after removing SNPs with absolute effect sizes smaller than or equal to 0.05 in Wood et al.
DOI: https://doi.org/10.7554/eLife.39725.031

We also excluded all UKB-candidate SNPs found in Wood et al. with absolute effect sizes smaller than or equal to 0.01, and recomputed the scores using the UK Biobank effect sizes (*Appendix 2—figure 7*).

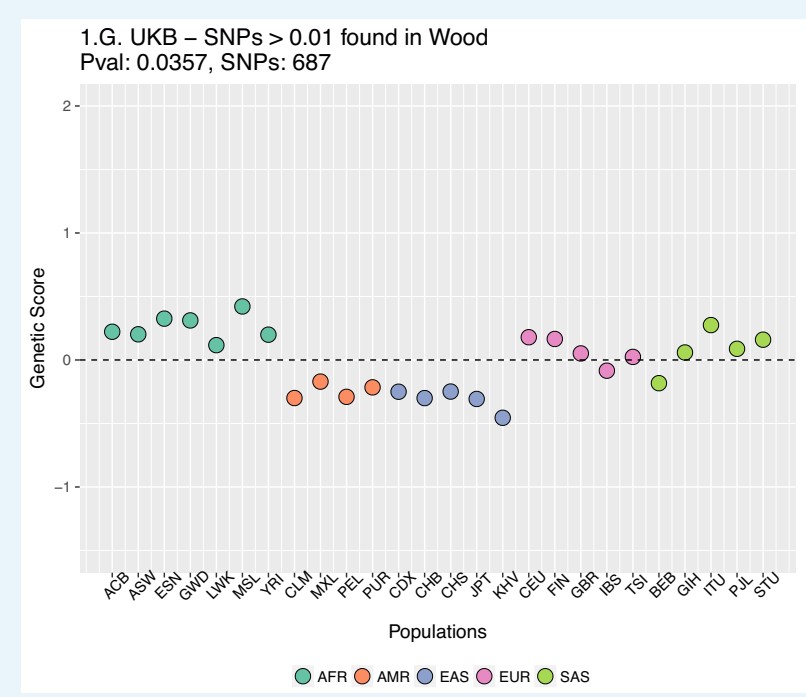

**Appendix 2—figure 7.** Genetic scores computed using the UK Biobank data, after removing SNPs with absolute effect sizes smaller than or equal to 0.01 in Wood et al.
DOI: https://doi.org/10.7554/eLife.39725.032

## Appendix 3

DOI: https://doi.org/10.7554/eLife.39725.018

## LD Score regression and linked selection

In this section we discuss how linked selection, specifically background selection (BGS), may be a potential confounder of LD Score regression. In the first section we discuss the intuition behind univariate LD Score regression and how BGS can cause a correlation between LD Score and allele frequency differentiation. In the second section we show empirically how LD Score and BGS covary across the genome, and how this can account for the empirical patterns of LD Score correlating with allele frequency differentiation. In the third section we show the BGS confounding of the slope and intercept of the univariate LD Score regression. In the final section we work through bivariate LD Score regression and show that it can be used to highlight the confounding of GWAS by specific axes of population structure.

Through this supplement we discuss the potential issue with linked selection in terms of BGS. However, it is likely that basic intuition of theses results, that is that linked selection is confounder of LD Score regression, apply more generally to other models of linked selection (e.g. selective sweeps).

### Background

**Bulik-Sullivan et al. (2015a)** and **Bulik-Sullivan et al. (2015b)** introduced LD Score regression as a robust way to assess the impact of population structure confounding on GWAS, and to robustly assess heritabilities and genetic correlations in GWAS even in the presence of such confounding. The LD Score of a SNP ($i$) is found by summing up LD ($R^2$) in a genomic window of $W$ surrounding SNPs:

$$\ell_i = \sum_{j=0}^{W} R_{i,j}^2. \tag{A9}$$

Following the logic laid out in the appendix of **Bulik-Sullivan et al. (2015b)**, consider a GWAS done using a sample drawn from two populations, with a sample of $N/2$ draws from each population. The trait is controlled by a very large number of loci ($M$), and the total narrow-sense heritability of the trait is $h_g^2$. The GWAS is partially confounded by population structure, as the squared difference in mean phenotype between the populations is $a$, and the allele frequency differentiation between the populations is $F_{ST}$. The expected $\chi_i^2$ statistic of the trait association of the $i^{th}$ SNP is

$$\mathbf{E}[\chi_i^2] = \frac{Nh_g^2}{M}\ell_i + 1 + aNF_{ST}, \tag{A10}$$

following Equation 2.7 of **Bulik-Sullivan et al. (2015b)**.

The basic idea of LD Score regression is that we regress $\chi_i^2$ on $\ell_i$, the deviation of the estimated intercept away from 1 gives $aNF_{ST}$, the confounding by population structure, while the slope of the regression gives $\frac{Nh_g^2}{M}$. Underlying this separation of the confounding effects of population structure ($aF_{ST}$) and the heritability ($h_g^2$) is the assumption that $F_{ST}$ is not correlated with LD Score. However, as noted by **Bulik-Sullivan et al. (2015b)** this assumption may be violated by background selection (BGS). In short, regions of low recombination (and thus higher LD Score) experience more BGS—which in turn drives higher $F_{ST}$ (**Charlesworth, 1998**).

To a first approximation, the effects of strong BGS in a well-mixed, constant-sized population can be modeled by a reduction in the effective population size, as the rate of drift increases in regions subject to BGS. We can express this mathematically by saying that SNP $i$ experiences an effective population size $B_iN_e$, where $N_e$ is the effective population size in the

absence of BGS and $B_i$ is the reduction due to BGS. The expected LD between SNP $i$ and another SNP $L$ $bp$ apart is

$$\mathbf{E}(R^2) \approx 1/(1 + 4N_e B_i r_{BP,i} L)$$

where $r_{BP,i}$ is the recombination rate surrounding SNP $i$.

$F_{ST}$, in turn, is a decreasing function of $N_e B_i$. For example, if the two populations at hand split $T$ generations ago, without subsequent gene-flow or population size changes,

$$\mathbf{E}(F_{ST}) \approx T/(4N_e B_i) \tag{A11}$$

(this approximation holds for small values of $T/N_e$). Similar inverse dependences of $F_{ST}$ on $B_i$ can be derived in other models of weak population structure (*Charlesworth, 1998*).

## Empirical results on LD Score and BGS

To explore the empirical relationship between LD Score, recombination rate and BGS we make use of the $B$ values estimated along the human genome by *McVicker et al. (2009)*. We use the 1000 Genomes CEU LD Scores (*Bulik-Sullivan et al., 2015b*), and the *Kong et al. (2010)* recombination rates (the latter are standardized by the genome-wide average recombination rate).

In *Figure 1* we plot the LD Score, averaged in 100 kb windows, as a function of recombination rates and McVicker's B values. As expected, LD Scores are higher in regions of low recombination and regions of stronger background selection (lower B). Based on a simple model of BGS (*Equation A11*), $F_{ST} \propto 1/B$. Therefore in *Figure 2* we plot the relationship between LD Scores and $1/B$ values each averaged in 30 quantiles of LD Score.

In the main text (*Figure 5A* and *Figure 5—figure supplement 1*) we plotted the relationship between LD Score and the $\chi^2$ statistic for allele frequency differentiation. To make our $\chi^2$ statistic comparable to $F_{ST}$ we standardized it. To do this we note that because population membership is not a genetic trait, setting $h^2 = 0$ in *Equation A10* we obtain

$$\mathbf{E}[\chi_i^2] = 1 + aNF_{ST}, \tag{A12}$$

Therefore, to make our $\chi_i^2$ statistic comparable to $F_{ST}$ we standardize our $\chi_i^2$ as:

$$(\chi_i^2 - 1)/\overline{\chi_i^2}, \tag{A13}$$

where the overbar in the denominator signifies a genome-wide average. In *Figure 2* we plot the expected relationship between LD Score and standardized $\chi_i^2$ predicted under our simple BGS model (using McVicker B values as an estimate for the intensity of background selection). We compare it to the the empirical relationship between LD Score and the standardized $\chi_i^2$ statistics for the Irish-British and GBR-TSI allele frequency differences. The agreement between the empirical results and the BGS-theoretical predictions is reasonable, suggesting that a model of BGS, as parameterized by McVicker's B, could explain the confounding in LD Score regression by linked selection.

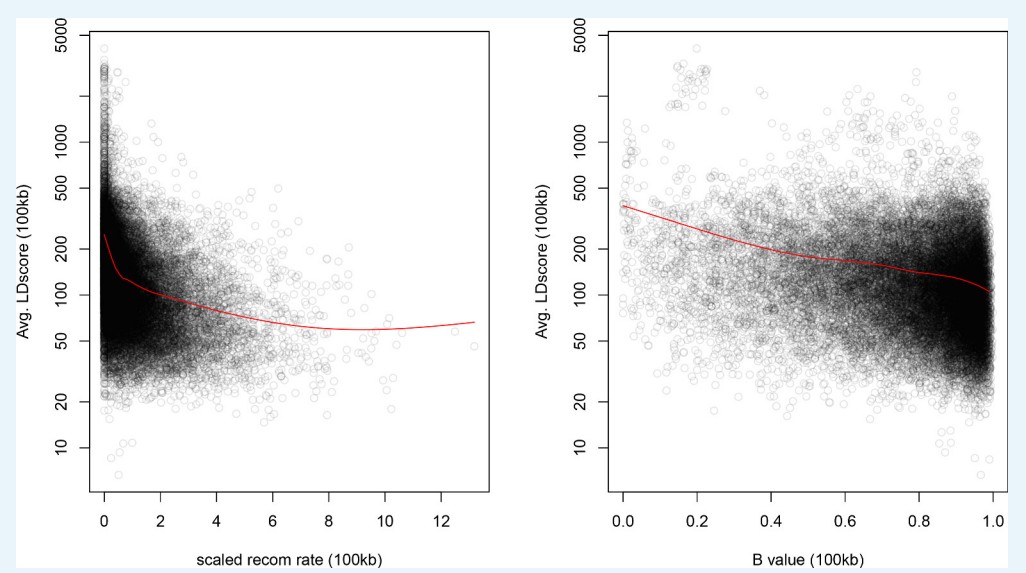

**Appendix 3—figure 1.** Windows with lower recombination rates and B values have higher LD Scores. The autosome is divided into 100 kb windows and the average LD Score, B-value, and standardized recombination rate is calculated in each bin. The red lines are a lowess fit as a guide to the eye.

DOI: https://doi.org/10.7554/eLife.39725.034

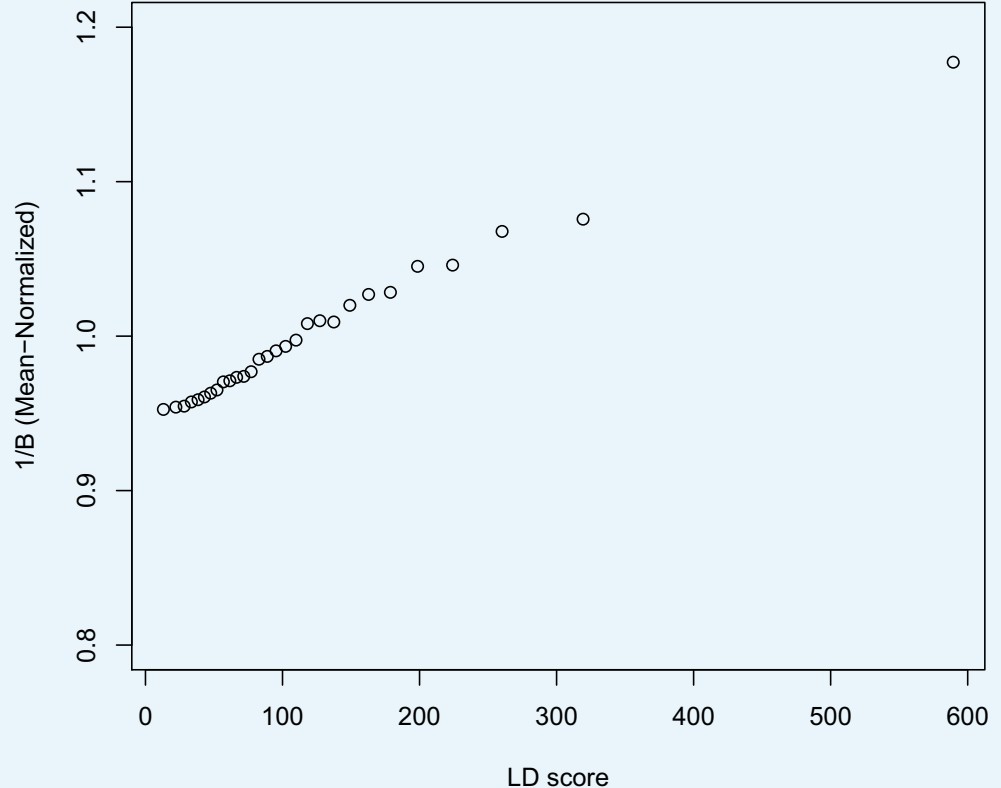

**Appendix 3—figure 2.** A plot across 30 quantiles of genome-wide LD Score of our simple BGS model of differentiation, parameterized by McVicker's B (*Equation A11*).

DOI: https://doi.org/10.7554/eLife.39725.035

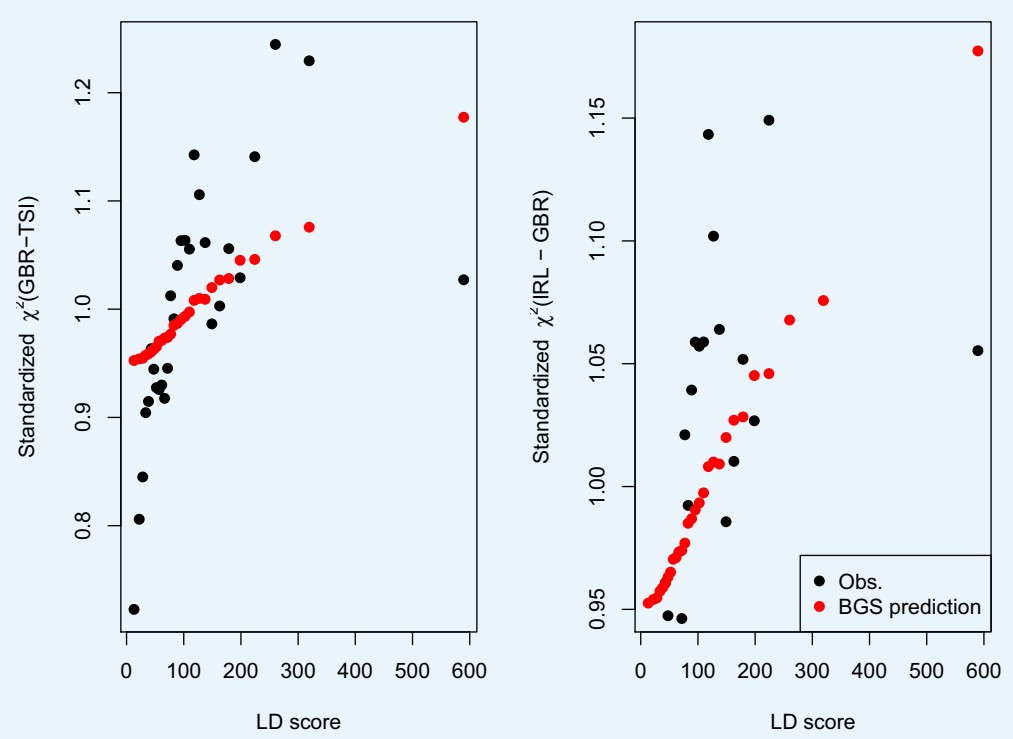

**Appendix 3—figure 3.** A plot across 30 quantiles of LD Score a standardized $\chi^2$ (*Equation A11*) of allele frequency differentiation (black dots) and that expected under our simple BGS model parameterized by McVicker's B (red dots, *Equation A11*, standardized by its genome-wide mean). Note that the red dots are the same values in both panels, and match those given in *Figure 2*.

DOI: https://doi.org/10.7554/eLife.39725.036

## Predicted effect on linked selection on the slope and intercept of LD Score regression.

The expectations of the slope and intercept of univariate LD Score were derived in the absence of linked selection. In this section we show how these expectations can be distorted by BGS.

In the regression of $\chi_i^2 \sim \ell_i$ the slope is:

$$\beta_{\chi^2;\ell} = \frac{Cov(\chi_i^2, \ell_i)}{Var(\ell_i))} \tag{A14}$$

$$= \frac{\frac{Nh_g^2}{M}Var(\ell_i) + aNCov(\ell_i, F_{ST,i})}{Var(\ell_i)} \tag{A15}$$

$$= \frac{Nh_g^2}{M} + aN\beta_{F_{ST};\ell} \tag{A16}$$

where $\beta_{F_{ST};\ell}$ is the slope of $F_{ST}$ regressed on LD Score. Therefore the slope of the univariate LD Scor regression is biased upwards by linked selection. The intercept is

$$\alpha_{\chi^2;\ell} = \overline{\chi^2} - \beta_{\chi^2;\ell}\overline{\ell} = 1 + aN(\overline{F_{ST}} - \beta_{F_{ST};\ell}\overline{\ell}), \tag{A17}$$

where the bars denote genome-wide averages. In other words, the intercept is suppressed by $aN\beta_{F_{ST};\ell}\overline{\ell}$. Another useful way to write the intercept is

$$\alpha_{\chi^2;\ell} = aN\overline{F_{ST}}(1 - \frac{\beta_{F_{ST};\ell}}{F_{ST}}\overline{\ell}), \tag{A18}$$

as $\frac{\beta_{F_{ST};\ell}}{F_{ST}}$ is the slope of the $\frac{F_{ST_i}}{F_{ST}} \sim \overline{\ell}_i$ regression—that is the effect of LD Score on the relative reduction in $F_{ST}$ from its mean.

## Using LD Score regression to assess 'genetic correlations' with allele frequency differentiation.

In the main text we plot the (Height GWAS effect size) × (Allele frequency difference) LD Score regression (*Figure 4D–F* and *Figure 4—figure supplement 3*). In a number of cases we see a strong intercept for this regression, and in some cases a significant slope. Here we show how a non-zero intercept may be a signal of stratification in the original GWAS along the axis represented by the allele frequency difference, while a non-zero slope may demonstrate that this stratification has interacted with BGS.

The logic of assessing genetic correlations via LD Score regression (*Bulik-Sullivan et al., 2015a*) is that at each SNP (*i*) we have a pair $(Z_{i,1}, Z_{i,2})$: scores for phenotypes 1 and 2 and the genetic correlation ($\rho_g$) between the phenotypes is captured by the slope of the regression $(Z_{i,1} \cdot Z_{i,2}) \sim \ell_i$. Imagine that these $Z$'s were estimated by conducting a GWAS of the two traits in a sample of size $N_1$ and $N_2$ respectively, with a sample overlap of $N_s$ individuals. The intercept of this regression, under the assumptions of *Bulik-Sullivan et al. (2015a)*, is determined by the phenotypic correlation ($\rho$) in the $N_S$ overlapping samples. *Bulik-Sullivan et al. (2015a)* show that under their assumptions of no stratification and no linked selection,

$$\mathbf{E}[Z_{i,1}Z_{i,2}] = \frac{\sqrt{N_1 N_2}\rho_g}{M}\ell_i + \frac{N_s\rho}{\sqrt{N_1 N_2}} \tag{A19}$$

*Yengo et al. (2018)* extended this to the case of a phenotype from a stratified population. Consider as before a population that consists of two equally sized samples from two populations with allele frequency differentiation $F_{ST}$. The difference in mean phenotype 1 and 2 between the two populations are $\sigma_1$ and $\sigma_2$ respectively. *Yengo et al. (2018)* show that

$$\mathbf{E}[Z_{i,1}Z_{i,2}] = \frac{\sqrt{N_1 N_2}\rho_g}{M}\ell_i + \frac{N_s\rho}{\sqrt{N_1 N_2}} + \rho_g F_{ST}^2\sqrt{N_1 N_2} + \frac{N_s^2 F_{ST}\sigma_1\sigma_2}{\sqrt{N_1 N_2}}. \tag{A20}$$

(This is equation (17) of *Yengo et al. (2018)*, up to slight differences in notation.)

Let us return to our case of the LD Score regression of (Height GWAS effect size) × (Allele frequency difference). Assume for the moment that our 'Allele frequency difference' (e.g. [GBR-TSI]) measures the difference in allele frequency between the two populations stratifying our GWAS. In our case, let phenotype 1 be a phenotype (e.g. height) and let 2 be an individual's population membership (e.g. 1 if in population 1 and 0 if in population 2) $Z_{i,H}$ and the $Z_{i,P}$ score-proxy of the allele frequency difference. The two phenotypes are measured in the same cohort (such that $N_1 = N_2 = N_S$. The difference in mean phenotype (height) between the two populations is $\sigma_1$. The mean difference in population membership is 1. As we can assume that population membership is not a genetic trait it follows that $\rho_g = 0$. However, there is a 'phenotypic' correlation between population membership and height, as height differs between our two populations stratifying our GWAS ($\rho = \sigma_1 \times 1$). Following the logic of *Equation A20* then

$$\mathbf{E}[Z_{i,H}Z_{i,P}] \approx A\sigma_1 + CF_{ST}\sigma_1 \tag{A21}$$

where A and C are constants. Note the strong similarity of *Equation A21* to the univariate LD Score regression for allele frequency $\chi^2$ (*Equation A12*). In reality the population samples (GBR and TSI) used to assess European N-S allele frequencies differences, in *Figure 4D–F*, and related figures, are not the population samples used in the GWAS. However, the spirit of

*Equation A21* holds if the confounding in a GWAS falls along this N-S axis. A significant intercept of this regression potentially indicates that some portion of the phenotypic variance (e.g. height) in the GWAS samples was confounded by residual N-S population structure and this problem has been transmitted through into the GWAS effect sizes. This LD Score regression is not necessarily expected to have any slope as *Equation A21* does not include the LD Score ($\ell_i$). However, if the population structure confounding ($F_{ST}$) in the GWAS samples is correlated with LD Score ($\ell_i$), for example due to BGS, then a slope will be induced (in a manner similar to *Equation A16*).

## Appendix 4

DOI: https://doi.org/10.7554/eLife.39725.018

### Testing QFAM's immunity to population stratification confounding

We set out to evaluate how effect sizes estimated in the sibling-based GWAS as implemented in plink's QFAM procedure are affected by population stratification. To this end, we added an artificial bias to the height of UK Biobank individuals along PC5-axis (we used PC5, among the top 40 PCs provided by the UK Biobank, along which the British ancestry individuals are most variable). We considered two cases. First, we added a bias proportional to the mean PC5 score in the family. Specifically, we set

$$Y_i^{fam-bias} = Y_i + \frac{2}{|F(i)|}\Sigma_{j\in F(i)}\gamma_j \tag{A22}$$

where $Y_i$ is an individuals actual recorded height in the UK Biobank, $F(i)$ indexes all siblings in individual $i$'s family, and $\gamma_j$ gives individual $j$'s projection onto PC5. This induced bias mimics an environmental contribution to the trait that varies with genetic ancestry across families but not within a family.

Second, we added a bias proportional to the individual's PC5 score:

$$Y_i^{ind-bias} = Y_i + 2\gamma_i. \tag{A23}$$

This mimics a scenario where there is a real genetic gradient in height along PC5. Height and PC5 score values were standardized as described in the section on newly calculated GWAS under Materials and methods.

In the first case, QFAM within-family effect size estimates are identical with and without including the bias, illustrating that plink v1.9b5's implementation correctly accounts for cross-family population structure (*Appendix 4—figure 1A*). Further, there is no correlation between the effect sizes and SNP loadings on PC5 (Figure *Appendix 4—figure 1B*, Pearson p = 0.81).

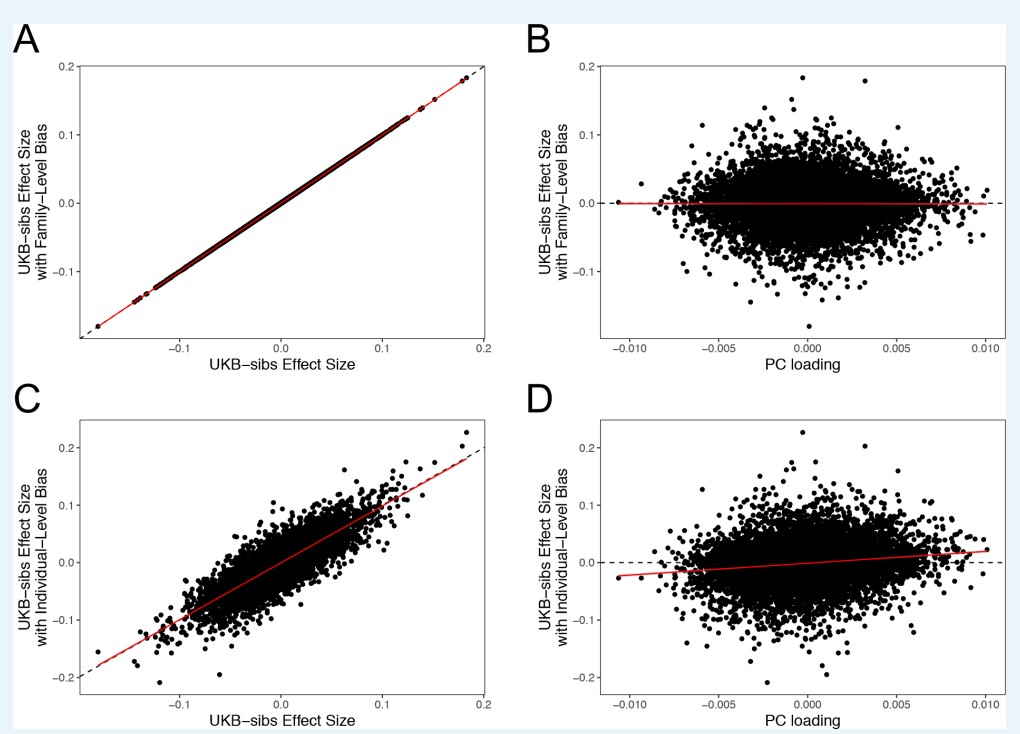

**Appendix 4—figure 1.** QFAM effect size estimates, under two population stratification scenarios. Top Row: Height values made biased along PC5-axis, proportional to the mean PC5 scores within family. (**A**) The x- and y-axes show effect size estimates without and with the added bias, respectively. (**B**) The x-axis shows the SNP loadings on PC5. Bottom Row: Height values made biased along PC5-axis, proportional to individuals' PC5 scores. (**C**) The same plot as panel (**A**), but with individual-level bias. (**D**) The same plot as panel (**B**), but with individual-level bias. All results are shown for 11,611 SNPs on chromosome one for which PC loadings where provided by the UK Biobank.
DOI: https://doi.org/10.7554/eLife.39725.038

In the second case, the within-family effect size estimates are biased (Figure ***Appendix 4—figure 1C***), proportional to the SNP contributions to PC5 (Figure ***Appendix 4—figure 1D***, Pearson p~$10^{-67}$). These results, however, do not reflect an issue with plink's implementation. Rather, they show that even correctly implemented family based studies can lead to biased effect size estimates if variation in ancestry segregates among siblings, provided that the different ancestries have different mean genetic contributions to the phenotype. There is no reason to think that this phenomenon is responsible for patterns seen in any of the real sibling datasets we analyze, but we present it here for completeness.

