## [Decision Letter]

[**Editorial note:** This article has been through an editorial process in which the authors decide how to respond to the issues raised during peer review. The Reviewing Editor's assessment is that all the issues have been addressed.]

Thank you for submitting your article "Reduced signal for polygenic adaptation of height in UK Biobank" for consideration by *eLife*. Your article has been reviewed by three peer reviewers, including Magnus Nordborg as the Reviewing Editor and Reviewer #1, and the evaluation has been overseen by Mark McCarthy as the Senior Editor. The following individuals involved in review of your submission have also agreed to reveal their identity: Nicholas H Barton (Reviewer #2) and Joachim Hermisson (Reviewer #3).

The Reviewing Editor has summarized the major concerns shared by all reviewers, and we have also included the separate reviews below for your consideration.

If you have any questions, please do not hesitate to contact us.

Summary:

This is one of two papers demonstrating that published signals of selection on human height cannot be replicated in the recently released UK Biobank data, apparently because these signals were caused by confounding population structure that is absent in UK Biobank data.

Major concerns:

We were struck by how both papers focus on spurious signals of selection rather than the underlying cause, which is that the GWAS effect-size estimates are confounded. The former is a somewhat esoteric question, but the latter may have enormous implications for much of human genetics, and these papers are likely to be heavily cited because of this. However, the papers seem to go out of their way to avoid discussing this topic. Of course we are not the authors, but, for the record, it looks odd.

Furthermore, the papers seem to suggest that confounding is not present in the UK Biobank data, but isn't it more likely that the magnitude is simply smaller?

The papers also present evidence that a sib-based study by Robinson et al., 2015, that was meant to eliminate confounding did no such thing. This is disturbing, and while we understand that identifying the reason may be beyond the present papers, the general implications should again probably be discussed.

Finally, while this is a carefully written and extremely scholarly paper, it relies heavily on population genetics jargon, and will be difficult to read for outsiders. We suggest substantial editing with this in mind.

Separate reviews (please respond to each point):

*Reviewer #1:*

This one of at least two papers appearing simultaneously and reaching exactly the same conclusion. It is well written, although perhaps a bit too technical for people not directly working in the field of human population genetics.

The only thing that surprises me about this paper is that it, as well as the other one I have seen, focuses on the relatively obscure issue of whether height has been under selection, tiptoeing around the much bigger issue (the elephant in the room) that the reason the claims for selection do not stand is that the GWAS estimates of effect sizes are biased because of population structure. It is not just the selection signals that do not replicate, but the polygenic scores. I'm not surprised, but, as you know, there are probably at least a hundred papers out there that are based on the infallibility of LD Score regression and genomic prediction. I understand the need for caution before attacking this edifice, but I nonetheless think some clarification is unavoidable.

*Reviewer #2:*

This is a carefully argued and scholarly examination of why estimates of selection, based on "polygenic scores", are weaker in the UK Biobank data than in earlier studies. This is an important issue, and so even though the causes of the discrepancy are not at all clear, the paper will be an important contribution. I have various comments below, and on the annotated PDF; several of these ask for a clearer definition of terminology that will make the paper more accessible to those not immersed in the field.

I would be encouraged by seeing a simulation of a stratified population, in which the standard correction for stratification fails, as is claimed to be the case here. One would also like to have some idea of what strength of allele frequency differentiation is needed to generate the observed differences in polygenic scores, and whether (as claimed) this indeed does not much affect GWAS estimates. However, this might be a non-trivial exercise, so I would not insist on it. I would like to see some discussion of how such a study could be approached, though.

The lesson from this is that it is extremely hard to reliably detect selection through the accumulation of slight changes in allele frequency. However, one should bear in mind that selection on polygenic traits is effective through precisely this process, and is largely unaffected by population structure. This is remarkable, but not paradoxical: natural selection can be effective even when its genetic basis cannot be identified.

Minor Comments:

Abstract: "polygenic differences netween populations" is perhaps ambiguous. You are not referring to differences in breeding value here, but to differences in BV inferred from SNP frequencies. Perhaps that can be said more directly.

Introduction first paragraph: Actually, polygenic scores predict an individual's additive genetic component, which is not the same as "individual phenotypes". Perhaps this could be explained: other studies of QG in the wild emphasise the importance of this distinction.

Subsection “GWAS data used to study adaptation of height” paragraph five: SDS only gets defined in the Table, it needs explanation in the text.

Subsection “Weak replication in UK Biobank” How confident can we that "current methods are likely sufficient for most applications" ? As noted above, I would have more confidence in this if we understood better how population structure confounds signals of polygenic adaptation.

Results section (and elsewhere): "genetic correlations" needs to be defined explicitly, correlations of what with what? Do these strong correlations imply that there is no detectable difference in effect size between populations? What bounds can one put on effect size differences

Figure 1: Best to label axes "Latitude, European" and "Longitude, Eurasian"

Subsection “SDS signal of selection in Britain.”: Define LMM! Since the correction for structure is crucial, it needs to be explained clearly early on.

Subsection “Relationship between GWAS estimates and European population structure”: Can one estimate the predicted strength of the "winner's curse"?

Subsection “LD Score regression signal”: "Meanwhile, confounders such as population structure are expected to affect SNPs of different LD Score equally": It is not obvious to me that this is the case.

Subsection “Summary of LD Score Regression Results”, final sentence: Delete "going forward"

Discussion section paragraph five: Nevertheless, one would like to know how much stratification may influence the accuracy of GWAS estimates.

Subsection “Expanded SNP Sets” final paragraph: One should be able to quantify this. We know the strength of assortment for height, and its effect on V_a_. Therefore, we can estimate its effect on pairwise LD, which will be very small because there are so may pairs, but may nevertheless lead to some overall bias. More directly, is there any correlation in polygenic scores between parents, consistent with the known assortment? One might be able to separate out causes of assortment in this way: stratification vs true assortment, causally due to height.

*Reviewer #3:*

Both manuscripts by Berg et al. and Sohail et al. present thorough and insightful analyses with highly relevant results for current and future GWAS studies. Even prior to publication, the manuscripts have considerable impact. They will be widely read and cited. I do not think that further analyses are needed, with the potential exception of the third point below. All other points concern the discussion, in particular the guidance for further research that will surely emerge from these studies.

# How safe are results based on the UK Biobank data?

This refers to the weak signals reported (with much caution) in the present studies, but also to potential future results on other traits. You recommend using data "such as UKB" and we will certainly see many more studies based on this resource. I would therefore appreciate a more specific discussion of risks connected to this particular data set.

1) Stratification even within the UKB-GB data: It is well known that height and socioeconomic status are correlated in modern societies (e.g. BMJ 2016; 352:i582), and social status correlates with descent. In the UK, both factors are also geographically stratified, with people living in the north of the country having lower socioeconomic status and shorter stature, on average, than those in the south. Furthermore, the percentage of Anglo-Saxon admixture varies across the UK. How could these factors influence results based on UKB data, both here and otherwise?

2) Potential influence of GxE interactions: The manuscripts focus (for good reason) on issues connected with stratification. However, if polygenic scores depend on the environment (e.g., due to countergradient variation), GxE interactions are an alternative confounding factor. Importantly, use of a homogeneous detection panel (to avoid stratification), such as UKB-GB, could increase these effects. Maybe this should be briefly discussed in the context of the present results and mentioned as a necessary caveat also for future studies that use detection panels from narrow geographic regions.

# What, exactly, causes the problems with the previous data?

3) There seem to be two relevant differences of the GIANT data relative to UKB: 1) UKB is much more homogeneous and 2) GIANT is a meta-study, collecting summary statistics from many sources that are individually corrected for stratification. One would like to know better which factor is decisive. This could be further addressed by combining summaries from sub-samples of the "UKB-all" data in an artificial meta-study.

4) The Robinson et al., 2015 GWAS: Sib-based studies are done to avoid/minimize stratification effects and the Robinson 2015 data have been used as a proof of robustness in several previous studies. The fact that you find clear signs of stratification is sobering and one would like to know what has gone wrong. You may not currently have any explanation and this is fair enough. However, the discussion should be clearer and say upfront that results based on these data cannot be trusted until we understand the issues.

Minor Comments:

a) The very low r^2^ in Figure 3C is strange. Any idea?

b) Figure 4: I was looking for the analysis using the other GB datasets, turns out that this is done in the supplementary information, but you should briefly comment on the results in the results part of the main text.

Additional data files and statistical comments:

Everything is well explained and all newly generated GWAS data are on Dryad.

---

## [Author Response]

Major concerns:We were struck by how both papers focus on spurious signals of selection rather than the underlying cause, which is that the GWAS effect-size estimates are confounded. The former is a somewhat esoteric question, but the latter may have enormous implications for much of human genetics, and these papers are likely to be heavily cited because of this. However, the papers seem to go out of their way to avoid discussing this topic. Of course we are not the authors, but, for the record, it looks odd.

One issue for us in writing this paper is that the effect at individual SNPs is relatively small, but systematic, therefore it is unclear how strongly various other types of analyses based on GWAS will be affected. This uncertainty means that in laying out the implications in a reasonable way we erred toward being conservative. We have now expanded our discussion to more broadly discuss the implications for polygenic score predictions, beyond just the spurious signals of selection, where the implications are clearer.

Furthermore, the papers seem to suggest that confounding is not present in the UK Biobank data, but isn't it more likely that the magnitude is simply smaller?

As discussed below we now point to evidence of residual confounding in the biobank.

The papers also present evidence that a sib-based study by Robinson et al., 2015, that was meant to eliminate confounding did no such thing. This is disturbing, and while we understand that identifying the reason may be beyond the present papers, the general implications should again probably be discussed.

Two weeks ago, Robinson and colleagues informed us that the effect sizes they released in 2016 had serious stratification issues due to a bug (see statement posted here: http://cnsgenomics.com/data.html). Furthermore, they revealed that these effect sizes that they released associated with their 2015 paper, were not in fact the effect sizes they used their 2015 paper. They also released a third set of effect sizes produced by reanalyzing the sibling data. This news obviously came to us late in our revisions process, but we have now run their new 2018 set of effect sizes through our pipeline. We now see no consistent signal of selection in Robinson et al’s new set of effect sizes, consistent with our BioBank analyses. Thus at least one mystery has now been solved.

Finally, while this is a carefully written and extremely scholarly paper, it relies heavily on population genetics jargon, and will be difficult to read for outsiders. We suggest substantial editing with this in mind.

We have substantially edited the paper to define terms, and make sure that it is more accessible.

Separate reviews (please respond to each point):

Reviewer #1:

This one of at least two papers appearing simultaneously and reaching exactly the same conclusion. It is well written, although perhaps a bit too technical for people not directly working in the field of human population genetics.The only thing that surprises me about this paper is that it, as well as the other one I have seen, focuses on the relatively obscure issue of whether height has been under selection, tiptoeing around the much bigger issue (the elephant in the room) that the reason the claims for selection do not stand is that the GWAS estimates of effect sizes are biased because of population structure. It is not just the selection signals that do not replicate, but the polygenic scores. I'm not surprised, but, as you know, there are probably at least a hundred papers out there that are based on the infallibility of LD Score regression and genomic prediction. I understand the need for caution before attacking this edifice, but I nonetheless think some clarification is unavoidable.

We have now extended our discussion of the issues for polygenic scores in the discussion, and how stratification is a strong source of false among-population variance for studies for samples with variation along axes of European variation.

Reviewer #2:

This is a carefully argued and scholarly examination of why estimates of selection, based on "polygenic scores", are weaker in the UK Biobank data than in earlier studies. This is an important issue, and so even though the causes of the discrepancy are not at all clear, the paper will be an important contribution. I have various comments below, and on the annotated PDF, several of these ask for a clearer definition of terminology that will make the paper more accessible to those not immersed in the field.

We have incorporated the reviewer’s comments about clarifying terminology throughout the document.

I would be encouraged by seeing a simulation of a stratified population, in which the standard correction for stratification fails, as is claimed to be the case here. One would also like to have some idea of what strength of allele frequency differentiation is needed to generate the observed differences in polygenic scores, and whether (as claimed) this indeed does not much affect GWAS estimates.. However, this might be a non-trivial exercise, so I would not insist on it. I would like to see some discussion of how such a study could be approached, though.

We agree that it would be useful to offer some simulations to show how stratification affected the original GWAS. However, the issue is that the source of population stratification in the meta-analysis is far from clear, as it involves many studies, each of which performed some level of correction for stratification. As such we feel that any simulation would not really provide much intuition about the level of confounding involved in the original GWAS.

The lesson from this is that it is extremely hard to reliably detect selection through the accumulation of slight changes in allele frequency. However, one should bear in mind that selection on polygenic traits is effective through precisely this process, and is largely unaffected by population structure. This is remarkable, but not paradoxical: natural selection can be effective even when its genetic basis cannot be identified.Minor Comments:Abstract: "polygenic differences netween populations" is perhaps ambiguous. You are not referring to differences in breeding value here, but to differences in BV inferred from SNP frequencies. Perhaps that can be said more directly.

Changed to “differences in polygenic score between populations”

Introduction first paragraph: Actually, polygenic scores predict an individual's additive genetic component, which is not the same as "individual phenotypes". Perhaps this could be explained: other studies of QG in the wild emphasise the importance of this distinction.

Changed to “additive genetic component of individual phenotypes”

Subsection “GWAS data used to study adaptation of height” paragraph five: SDS only gets defined in the Table, it needs explanation in the text.

Fixed.

Subsection “Weak replication in UK Biobank”: How confident can we that "current methods are likely sufficient for most applications" ? As noted above, I would have more confidence in this if we understood better how population structure confounds signals of polygenic adaptation.

Changed “most” to “many”, and outlined a few contrasting cases where current methods seem likely sufficient, vs. those where they may not be.

Results section (and elsewhere): "genetic correlations" needs to be defined explicitly, correlations of what with what? Do these strong correlations imply that there is no detectable difference in effect size between populations? What bounds can one put on effect size differences

Added text explaining the procedure for calculating “genetic correlations”, and how we are interpreting them.

Figure 1: Best to label axes "Latitude, European" and "Longitude, Eurasian"

Done.

Subsection “SDS signal of selection in Britain.”: Define LMM! Since the correction for structure is crucial, it needs to be explained clearly early on.

Done.

Subsection “Relationship between GWAS estimates and European population structure”: Can one estimate the predicted strength of the "winner's curse"?

It is possible, but not trivial (e.g. https://www.sciencedirect.com/science/article/pii/S0002929707610970, https://onlinelibrary.wiley.com/doi/abs/10.1002/gepi.20398), to correct for the winner’s curse, but it’s unclear to us in our current application what the practical advantage of doing so would be.

Subsection “LD Score regression signal”: "Meanwhile, confounders such as population structure are expected to affect SNPs of different LD Score equally": It is not obvious to me that this is the case.

This expectation traces its origin to the original LD Score paper (Bulik Sullivan et al. 2015), which argued that the rate of drift and LD Score should be uncorrelated. They acknowledged that background selection could potentially violate this argument, and performed a few simulations in small samples of northern Europeans to support it. This assertion of independence between rate of drift and LD Score has been generally accepted by the human genetics community. However, as we show later in this paper, this assumption does not necessarily hold.

We replaced the word “expected” with “argued” to clarify that we do not ourselves necessarily expect this to hold, but that it has been argued.

Subsection “Summary of LD Score Regression Results”, final sentence: Delete "going forward"

Done.

Discussion section paragraph five: Nevertheless, one would like to know how much stratification may influence the accuracy of GWAS estimates.

We have updated this section of the paper to more thoroughly discuss our current state of knowledge.

Subsection “Expanded SNP Sets” final paragraph: One should be able to quantify this. We know the strength of assortment for height, and its effect on V_a_. Therefore, we can estimate its effect on pairwise LD, which will be very small because there are so may pairs, but may nevertheless lead to some overall bias. More directly, is there any correlation in polygenic scores between parents, consistent with the known assortment? One might be able to separate out causes of assortment in this way: stratification vs true assortment, causally due to height.

This is an interesting idea that we think may be worth pursuing in the future. However, we note that the accelerated rate of drift relative to our neutral expectation depends on the degree of assortative mating in the past, when the specified drift was taking place, and may therefore differ from the rate that would be expected given the present day degree of assortative mating. The best that can likely be estimated from this sort of signature is some sort of time averaged rate of assortative mating over the period during which the populations under study were diverging.

This does raise the point that, so long as stabilizing selection is not too strong, strong assortative mating over short periods of time may be essentially indistinguishable from bouts of selection. This is a topic some of us are interested in exploring in the future.

Reviewer #3:

Both manuscripts by Berg et al. and Sohail et al. present thorough and insightful analyses with highly relevant results for current and future GWAS studies. Even prior to publication, the manuscripts have considerable impact. They will be widely read and cited. I do not think that further analyses are needed, with the potential exception of the third point below. All other points concern the discussion, in particular the guidance for further research that will surely emerge from these studies.# How safe are results based on the UK Biobank data?This refers to the weak signals reported (with much caution) in the present studies, but also to potential future results on other traits. You recommend using data "such as UKB" and we will certainly see many more studies based on this resource. I would therefore appreciate a more specific discussion of risks connected to this particular data set.1) Stratification even within the UKB-GB data: It is well known that height and socioeconomic status are correlated in modern societies (e.g. BMJ 2016; 352:i582), and social status correlates with descent. In the UK, both factors are also geographically stratified, with people living in the north of the country having lower socioeconomic status and shorter stature, on average, than those in the south. Furthermore, the percentage of Anglo-Saxon admixture varies across the UK. How could these factors influence results based on UKB data, both here and otherwise?

We now have a brief discussion of remaining stratification in the UK Biobank, and point to other papers on this topic.

2) Potential influence of GxE interactions: The manuscripts focus (for good reason) on issues connected with stratification. However, if polygenic scores depend on the environment (e.g., due to countergradient variation), GxE interactions are an alternative confounding factor. Importantly, use of a homogeneous detection panel (to avoid stratification), such as UKB-GB, could increase these effects. Maybe this should be briefly discussed in the context of the present results and mentioned as a necessary caveat also for future studies that use detection panels from narrow geographic regions.

This is a fair point, but also one that has been covered fairly extensively in the Berg and Coop 2014 paper, as well as the recent 2018 commentary from Novembre and Barton. We now include a comment regarding the promises and pitfalls of studying polygenic phenotypes across ancestry and environmental gradients in our closing paragraph, and direct readers to these two papers.

# What, exactly, causes the problems with the previous data?3) There seem to be two relevant differences of the GIANT data relative to UKB: 1) UKB is much more homogeneous and 2) GIANT is a meta-study, collecting summary statistics from many sources that are individually corrected for stratification. One would like to know better which factor is decisive. This could be further addressed by combining summaries from sub-samples of the "UKB-all" data in an artificial meta-study.

We agree this would be interesting. However, a thorough investigation of this issue could easily constitute an entire paper of its own, and without access to the original GIANT cohort data, would bring us no closer to determining what actually happened in that dataset.

4) The Robinson et al., 2015 GWAS: Sib-based studies are done to avoid/minimize stratification effects and the Robinson 2015 data have been used as a proof of robustness in several previous studies. The fact that you find clear signs of stratification is sobering and one would like to know what has gone wrong. You may not currently have any explanation and this is fair enough. However, the discussion should be clearer and say upfront that results based on these data cannot be trusted until we understand the issues.

Robinson et al. have now informed us that the summary statistics they posted online in 2016 were A) not the same statistics actually used in their 2015 paper, and B) afflicted by a plink bug which resulted in mishandling of family structured data, and therefore no control for population structure (http://cnsgenomics.com/data.html). We have updated the paper throughout to reflect this, and we added an analysis of a new set of corrected summary statistics they released.

Minor Comments:a) The very low r^2^ in Figure 3C is strange. Any idea?

This is due to the substantially lower power of the Robinson sib study relative to GIANT. This should result in a greater distance, on average, between the chosen lead SNP and the causal variation it tags, leading to much weaker replication of effect sizes. We have added a comment on the difference in power to the figure caption.

b) Figure 4: I was looking for the analysis using the other GB datasets, turns out that this is done in the supplementary information, but you should briefly comment on the results in the results part of the main text

Added.